



# Measurement report: Contribution of atmospheric new particle formation to ultrafine particle concentration, cloud condensation nuclei and radiative forcing: Results from five-year observations in Central Europe

Jia Sun[1,2], Markus Hermann[2], Kay Weinhold[2], Maik Merkel[2], Wolfram Birmili[3], Yifan Yang[2], Thomas Tuch[2], Harald Flentje[4], Björn Briel[4], Ludwig Ries[3], Cedric Couret[3], Michael Elsasser[3], Ralf Sohmer[3], Klaus Wirtz[3], Frank Meinhardt[3], Maik Schütze[3], Olaf Bath[3], Bryan Hellack[3], Veli-Matti Kerminen[5], Markku Kulmala[5], Nan Ma[6], Alfred Wiedensohler[2]

[1]Southern Marine Science and Engineering Guangdong Laboratory (Guangzhou), Guangzhou, China
[2]Leibniz Institute for Tropospheric Research (TROPOS), Leipzig, Germany
[3]German Environment Agency (UBA), Berlin, Germany
[4]Deutscher Wetterdienst (DWD), Meteorologisches Observatorium Hohenpeißenberg, Germany
[5]Department of Physics, University of Helsinki, P.O. Box 64, 00014 Helsinki, Finland
[6]Institute for Environmental and Climate Research, Jinan University, Guangzhou, Guangdong 511443, China

*Correspondence to*: Nan Ma (nan.ma@jnu.edu.cn) and Alfred Wiedensohler (ali@tropos.de)

**Abstract.** As an important source of sub-micrometer particles, atmospheric new particle formation (NPF) has been observed in various environments. However, most studies provide little more than snapshots of the NPF process due to their underlying observations being limited in space and time. To obtain statistically relevant evidence on NPF across various environments, we investigated the characteristics of NPF based on a five-year dataset of the German Ultrafine Aerosol Network (GUAN). The results were also compared with the observations in previous studies, aiming to depict a relatively complete picture of NPF in Central Europe. The highest NPF frequency was observed in regional background, with an average of about 20%, followed by urban background (15%), low mountain range (8%) and high Alpine (3%). The annual mean growth rate varied from 3.67 to 4.70 nm h$^{-1}$, while the formation rate from 0.43 to 2.89 cm$^{-3}$ s$^{-1}$. The contribution of NPF on UFPs was about 13%, 21%, and 7% for the urban background, regional background, and low mountain range, respectively. The influence of NPF on CCN number concentration and aerosol extinction coefficient for NPF days were the highest in mountainous area. These findings underscore the importance of the local environments when assessing the potential impact of NPF on regional climate in models, and also emphasize the usefulness of a long-term aerosol measurement network for understanding the variation of NPF features and their influencing factors over a regional scale.





## 1 Introduction

Atmospheric new particle formation (NPF) is a process that creates new aerosol particles smaller than 3 nm in diameter from gas-phase precursors. These particles may subsequently grow into larger sizes by condensation or coagulation (Kulmala et al., 2014), resulting in a growing number concentration of ultrafine particles (UFPs, particles smaller than 100 nm) or even larger sub-micrometer particles (particles smaller than 1 μm) (Ma and Birmili, 2015). Once the newly formed particles grow into
larger sizes (typically more than 100 nm), they can affect cloud properties and processes by acting as cloud condensation nuclei (CCN) (Dameto De España et al., 2017; Hirshorn et al., 2022; Ren et al., 2021). As an essential source of atmospheric aerosols, NPF events can also impact the regional radiative forcing of the atmosphere by increasing the overall extinction of light as the particles grow larger (Shen et al., 2011).

NPF is a complex process affected by various factors, including meteorological conditions (Bousiotis et al., 2021a; Li et
al., 2019; Salvador et al., 2021), atmospheric chemical composition (Dada et al., 2020; Dall'Osto et al., 2018; Németh et al., 2018; Nieminen et al., 2014), and pre-existing aerosol loading (Bousiotis et al., 2021b; Salma and Németh, 2019). Studies on NPF have been conducted in diverse environments, ranging from polluted megacity (Yao et al., 2018; Wang et al., 2014) to clean areas (Petäjä et al., 2009; Vana et al., 2016). Experimental observations show that in the continental boundary layer, NPF often occurs in the shape of "NPF events", i.e., the nucleation and subsequent growth of particles may take place over
horizontal spatial scales up to several tens or hundreds of kilometers. Such "banana" type NPF events with particle formation and growth can accordingly occur across various locations and diverse types of environments (Kerminen et al., 2018). The basis of experimental observations has grown steadily over the past 30 years, and a plethora of computational models have been developed to describe NPF on mechanistic and empirical levels.

However, the conclusions drawn from existing studies show large discrepancies, and the influence of local atmospheric or
meteorological conditions on NPF has not been fully understood yet. For instance, although some studies reported that low ambient relative humidity (RH) environments favour NPF (Cai et al., 2017; Dada et al., 2017; Li et al., 2019), NPF has still been observed in the environments with high RH (O'Dowd et al., 1998; Bousiotis et al., 2021b). High temperature associated with strong solar radiation can promote photochemical reaction and nucleation (Boy and Kulmala, 2002; Kürten et al., 2016; Ma and Birmili, 2015) and a well-mixed air leads to low condensational sink (CS), resulting in a higher probability for NPF
(Größ et al., 2018; Dall'Osto et al., 2018; Bousiotis et al., 2021a). Conversely, high temperature and a well-mixed atmosphere may also inhibit NPF by decreasing the stability of molecular clusters (Hanson et al., 2017; Kürten et al., 2018). Additionally, the role of mixed atmospheric chemical species, including $SO_2$, $NH_3$, and volatile organic compounds (VOCs), is complex and varies with the nucleation mechanism and concentration of those components (Laaksonen et al., 2008; Ehn et al., 2014; Kürten et al., 2016; Qi et al., 2018). Existing theories still cannot fully explain the fundamental chemical mechanisms of NPF events
observed under diverse tropospheric environments and the result of field measurements are often controversial concerning the contribution of the chemical species to nucleation and growth of nanoparticles (Lee et al., 2019).



Continuous observations of NPF started as single point observations at ground level (Mäkelä et al., 1997; Birmili and Wiedensohler, 2000), and were subsequently expanded to cover greater spatial and temporal scales. Several studies have investigated NPF at multiple sites at small-region-scale (for example around a city) (Costabile et al., 2009; Németh and Salma, 2014; Bousiotis et al., 2019; Casquero-Vera et al., 2020; Kalkavouras et al., 2020; Smejkalova et al., 2021; etc.), at country- or continent-scale (Manninen et al., 2010; Dall'Osto et al., 2018; Németh et al., 2018; Bousiotis et al., 2021a; Sebastian et al., 2022; etc.), and at global-scale (Ren et al., 2021; Nieminen et al., 2018; Sellegri et al., 2019). Small-region-scale studies refer to individual NPF observations regional and closer distance (< 200 km), such as in central France (Boulon et al., 2011), Budapest (Németh and Salma, 2014; Salma et al., 2017), southern UK (Bousiotis et al., 2019), and Leipzig (Ma and Birmili, 2015). These studies mainly focused on the difference in NPF features with varied degree of anthropogenic and biogenic emissions (Ma and Birmili, 2015; Bousiotis et al., 2019), or the characteristics of NPF events occurring simultaneously at several sites in a small area (Németh and Salma, 2014; Salma et al., 2016). Country-based or continental-scale studies can provide insight into the connection between NPF events and their influencing factors covering a particular region, such as Europe (Dall'Osto et al., 2018; Bousiotis et al., 2021b; Manninen et al., 2010) and India (Sebastian et al., 2022). Global analyses of NPF events stretch these comparisons further, comparing the characteristics of NPF under one (Sellegri et al., 2019) or multiple types of environments (Nieminen et al., 2018; Ren et al., 2021). However, comprehensive observations of regional NPF across multiple sites have still been limited to date, with most studies including only 2 or 3 sites, leading to open questions when explaining the spatial and temporal variabilities of regional NPF across diverse environments throughout a large region.

The German Ultrafine Aerosol Network (GUAN) is an observation network for sub-micrometer aerosol particles measurements, aiming at a better understanding of the associated climate and health effects. GUAN provides long-term atmospheric aerosol measurements in diverse site categories in Germany, ranging from roadside to high Alpine area (Birmili et al., 2016; Sun et al., 2020). On the basis of the GUAN observations, a comprehensive comparison of NPF in various environments across Germany becomes realistic. Based on a five-year dataset we investigated the characteristics of NPF for various environments from urban background to high Alpine, including the occurrence of NPF events, particle formation and growth rates, and the impacts of NPF on UFP, CCN and radiative forcing, aiming to depict a relatively complete picture of NPF in Central Europe.

## 2 Measurement and data

### 2.1 Measurement sites

This study uses atmospheric observations from nine observation sites in the German Ultrafine Aerosol Network (GUAN; Birmili et al., 2016; https://doi.org/10.5072/guan, last access: 30 August 2023). GUAN is a cooperative observation network of several research organizations providing continuous measurement of sub-micrometer particle number size distributions (PNSD) and equivalent black carbon (eBC) mass concentration since 2009. GUAN consists of 17 measurement sites covering diverse environmental settings in Germany including roadside, urban background, regional background, low mountain range





and high Alpine. The locations and characteristics of the nine selected measurement sites are shown in Fig.1 and Table 1. For

a detailed description, see Birmili et al. (2016).

The nine GUAN measurement sites in this study comprise three urban background sites, three regional background sites, two low mountain range sites and one high Alpine site. Three urban background sites are Leipzig-West (LWE), Leibniz Institute for Tropospheric Research (TROPOS) (LTR), and Bösel (BOS). LWE and LTR are both located in the city of Leipzig with 10 km apart. LTR is situated on the roof of the TROPOS main building, while LWE is settled in a hospital park in the

western suburbs of Leipzig. BOS is located in the village of Bösel, about 100 km away from the North Sea.

The regional background site Melpitz (MEL) is distant about 50 km in the north-east of Leipzig. Its surroundings of MEL are flat and seminatural grasslands without significant anthropogenic sources. Site Neuglobsow (NEU) and Waldhof (WAL) are situated in Northern Germany forest regions. One previous study showed that MEL can represent the regional background atmosphere of Central Europe (Spindler et al., 2013), while NEU and WAL represent the regional background condition in the

northern Germany lowlands (Sun et al., 2019).

Three mountainous sites in GUAN are located in the southern Germany, including two low mountain range sites Schauinsland (SCH, 1205 m a.s.l.) and Hohenpeißenberg (HPB, 980 m a.s.l.), and one high Alpine site Zugspitze-Schneefernerhaus (ZSF, 2670 m a.s.l.). SCH is situated in the Black Forest and HPB is on a solitary hill in the countryside of southern Bavaria, 40 km north of the Alpine Mountain range and 45 km southwest of Munich region. As a part of the World

Meteorological Organisation (WMO) Global Atmosphere Watch (GAW) program, ZSF is located on the south side of Zugspitze Mountain approximately 300 m below the summit and the airmasses of both lower free troposphere (FT) and planetary boundary layer (PBL) can be observed there (Sun et al., 2021; Yuan et al., 2019).

## 2.2 Instrumentation

Aerosol PNSD was measured by either Mobility Particle Size Spectrometers (MPSS, Wiedensohler et al., 2012), dual mobility

particle size spectrometers (D-MPSS). Some stations used an additional thermodenuder option, Thermodenuder Mobility Particle Size Spectrometers (TDMPSS, Wang et al., 2017), whose data, however, were not used in this analysis. The specifications of the instruments used at each site are summarized in Table 1. To ensure standardized conditions for particle sizing at different sites and times, PNSD were generally measured in a dry state with RH below 40 % (Swietlicki et al., 2008). An inversion algorithm developed by Pfeifer et al. (2014) based on bipolar charge distribution (Wiedensohler, 1988) was used

to retrieve the PNSD from the measured raw mobility distribution. The particle losses in instruments and inlet systems were corrected based on Wiedensohler et al. (2012) and the quality assurance (QA) were done as described in Wiedensohler et al. (2018).

The QA of MPSS measurements in GUAN, including both instrument to instrument and instrument to standard comparisons, was regularly conducted by the World Calibration Centre for Aerosol Physics (WCCAP, http://www.wmo-gaw-

wcc-aerosol-physics.org/, last access:12 April 2023) in Leipzig. The aim of QA is to obtain an accuracy within a few percent for the particle sizing and ±10 % for particle number concentration (PNC) of PNSD over the entire measurement period. The



periodical QA procedures for MPSS includes daily or weekly inspection, monthly and annual full maintenance, either at measurement site or at laboratory of WCCAP. Detailed descriptions of the QA procedure are given in Birmili et al. (2016).

The PNSD data used in this study covers a five-year period from 2009 to 2013, with three exceptions: NEU and LWE
started PNSD measurements in 2011 and the PNSD data at ZSF are available from 2012. The temporal coverages of qualitied PNSD data at the nine sites are given in Fig.S1 in the supplementary material.

## 2.3 Method

### 2.3.1 NPF events classification

The classification of NPF events was performed visually according to the criteria given by Dal Maso et al. (2005). If a distinct
new nucleation mode (3−25 nm) appeared and grew into the Aitken mode size range (25−100 nm) within the subsequent hours between 00:00 and 24:00 local time, such a day was classified as a NPF day. The NPF event was classified as type I if the formation and growth rate of the NPF event could be clearly determined from the observed evolution of the PNSD, and type II if not. The formation and growth rates were calculated only for type I events. Type I events were further grouped into two sub-class: Ia and Ib. Class Ia contains very strong and clear NPF with "banana shape". And the rest of type I events were
classified as class Ib. The days were classified as "undefined event" if the cases could not be clearly classified as event or non-event.

### 2.3.2 Calculation of growth and formation rates

The growth rate of nucleation mode particles $N_{10-25}$ (GR$_{nuc}$) is defined as the change rate of the mean diameter of the newly formed particles (Kulmala et al., 2012): Eq. (1):

$$GR_{nuc} = \left.(D_{P_2} - D_{P_1})\middle/(t_2 - t_1)\right. , \tag{1}$$

where $D_{P_1}$ and $D_{P_2}$ are the geometric mean diameters (GMDs) of the mode of newly formed particles at starting and ending time during a NPF event. The GMDs were obtained by the log-normal modal fitting of the PNSD.

Formation rates in nucleation mode ($J_{nuc}$) is the sum of the increase rate, decrease rate of $N_{10-25}$ due to coagulation loses, and decrease rate of $N_{10-25}$ due to the condensational growth out of the nucleation mode. Accordingly, $J_{nuc}$ was obtained using
the following equation (Kulmala et al., 2012):

$$J_{nuc} = \frac{dN_{10-25}}{dt} + CoagS_{D_p} \times N_{10-25} + \frac{GR_{nuc}}{\Delta D_p} \times N_{10-25} \tag{2}$$

where $CoagS_{D_p}$ is the coagulation sink of particles with diameter $D_p$, which can be calculated using the method proposed by Kerminen et al. (2001):

$$CoagS_{D_p} = \sum_{D'_p=D_p}^{D'_p=max} K(D_p, D'_p) N_{D_p} \tag{3}$$



where $K(D_{\mathrm{p}}, D_{\mathrm{p}}')$ is the coagulation coefficient of particle size $D_{\mathrm{p}}$ and $D_{\mathrm{p}}'$, and $N_{D_{\mathrm{p}}}$ is the particle number concentration of

particle with size $D_{\mathrm{p}}$.

### 2.3.3 Nucleation strength factor

The nucleation strength factor (NSF) proposed by Németh and Salma (2014) qualitatively evaluates the overall concentration

increment on NPF days exclusively, calculated by:

$$\mathrm{NSF}_{\mathrm{nuc}} = \frac{\left(N_{10-100}/N_{100-800}\right)_{\mathrm{all\ NPF\ event\ days}}}{\left(N_{10-100}/N_{100-800}\right)_{\mathrm{all\ non-event\ days}}} \tag{4}$$

### 2.3.4 Contribution to UFP number concentration

The contribution of NPF on UFP number concentration was quantitively estimated by segregating the diurnal patterns of UFP

driven by NPF, urban sources and regional background (Ma and Birmili, 2015). As observed in the latter reference, NPF events

occurred almost exclusively on days with a daily average solar radiation of more than 100 W m$^{-2}$ in Germany. Subsequently,

the measurement period was firstly separated into high and low solar radiation days by a threshold of daily average solar

radiation 100 W m$^{-2}$, to accurately estimate the effect of NPF to UFP number concentration. The average diurnal cycles of

UFP number concentration for NPF days and non-event days at high solar radiation period were calculated and denoted as

$\widetilde{N}_{HR-NPF}$ and $\widetilde{N}_{HR-NON}$, respectively. Similarly, the corresponding values at low solar radiation period were calculated and

denoted as $\widetilde{N}_{LR-NPF}$ and $\widetilde{N}_{LR-NON}$. The average number concentration of newly formed particles for high and low radiation

days were respectively calculated as:

$$\bar{N}_{\mathrm{NPF-HR}} = \frac{\int_0^{24}(\widetilde{N}_{\mathrm{HR-NPF}} - \widetilde{N}_{\mathrm{HR-NON}}) \times \mathrm{d}t}{24} \tag{5}$$

$$\bar{N}_{\mathrm{NPF-LR}} = \frac{\int_0^{24}(\widetilde{N}_{\mathrm{LR-NPF}} - \widetilde{N}_{\mathrm{LR-NON}}) \times \mathrm{d}t}{24} \tag{6}$$

Accordingly, the overall contribution of NPF event to UFP concentration can be calculated as:

$$\bar{N}_{\mathrm{NPF}} = \frac{\bar{N}_{\mathrm{NPF-HR}} \times n_{\mathrm{NPF-HR}} + \bar{N}_{\mathrm{NPF-LR}} \times n_{\mathrm{NPF-LR}}}{n_{\mathrm{NPF-HR}} + n_{\mathrm{NPF-LR}} + n_{\mathrm{NON-HR}} + n_{\mathrm{NON-LR}}} \tag{7}$$

where $n_{\mathrm{NPF-HR}}$, $n_{\mathrm{NPF-LR}}$, $n_{\mathrm{NON-HR}}$, and $n_{\mathrm{NON-LR}}$ are the number of days with high/low radiation and with/without NPF

events, respectively.

### 2.3.5 Enhancement in CCN number concentration

The NPF-initiated enhancements in CCN number concentration ($N_{\mathrm{CCN}}$) enhancement, denoted as $E_{N\mathrm{ccn}}$, was quantified using

the method proposed by Ren et al. (2021) and Kalkavouras et al. (2019). This approach compares the $N_{\mathrm{CCN}}$ between after and

prior to the NPF event:

$$E_{N\mathrm{ccn}} = N_{\mathrm{CCN\_after}} \Big/ N_{\mathrm{CCN\_prior}} \tag{8}$$





where $N_{\text{CCN\_prior}}$ is the two-hour average of $N_{\text{CCN}}$ before the start of NPF, and $N_{\text{CCN\_after}}$ is determined as the average $N_{\text{CCN}}$ during the period that NPF contributing on $N_{\text{CCN}}$. As a simplified estimate, $N_{\text{CCN}}$ was calculated as the integral PNC with particle size larger than pre-defined critical diameter ($D_{\text{C}}$). Referring to a previous study by Wu et al. (2015), $D_{\text{C}}$ of 50 nm, 70 nm and 180 nm were applied for 0.6, 0.4, and 0.1 % supersaturation, respectively. The start and end time of the period that NPF impacts on $N_{\text{CCN}}$ were determined by evaluating the variability of normalized time series of $N_{\text{CCN}}$ at each prescribed supersaturation. The detailed approach can be seen in Kalkavouras et al. (2019) and Ren et al. (2021). It should be noted that this method is based on the assumption that the background concentration of CCN holds constant during the NPF, ignoring the influence of other sources and sinks of aerosol particles, therefore it can only give a rough estimate of the impact of NPF on $N_{\text{CCN}}$.

### 2.3.6 Enhancement in extinction coefficient

The influence of NPF on radiative forcing was evaluated based on the measured PNSD and eBC mass concentration using the Mie theory. Assuming that BC is internal mixed and its volume fraction is independent of particles size, a uniform volume fraction of BC ($VF_{\text{BC}}$) for different particle sizes is defined as

$$VF_{BC} = \bar{V}_{BC}/\overline{PVC} \tag{9}$$

where, $\bar{V}_{BC}$, the mean volume concentration of BC particles is obtained by the mean eBC mass concentration during the observation period divided by the density of BC (1.5 g/cm$^3$). $\overline{PVC}$ is the average integral particle volume concentration (PVC) calculated from the measured PNSD.

Accordingly, the refractive index is derived as a volume-weighted average of BC and non-absorbing component:

$$\bar{m} = VF_{\text{BC}} \times \bar{m}_{\text{BC}} + (1 - VF_{\text{BC}}) \times \bar{m}_{\text{non-abs}} \tag{10}$$

where the refractive index for BC is set as $\bar{m}_{\text{BC}} = 1.96 - 0.66i$ (Seinfeld et al., 1998), and for non-absorbing component $\bar{m}_{\text{non-abs}} = 1.53 - 10^{-7}i$ (Wex et al., 2002).

The dimensionless extinction efficiency $Q_{\text{ext}}$ can be obtained using Mie theory (Mie, 1908), and the extinction coefficient $\sigma_{\text{ext}}$ can be calculated accordingly as:

$$\sigma_{\text{ext}} = \int_{D_P} Q_{\text{ext}} \times (\frac{\pi}{4}D_P^2) \times \text{PNSD} \times d\log D_P \tag{11}$$

Similar as the CCN enhancement estimation in Sect.2.3.5, the NPF-initiated enhancement of aerosol extinction coefficient ($E_{\text{ext}}$) was quantified as:

$$E_{\text{ext}} = {\sigma_{\text{ext\_after}}}\big/{\sigma_{\text{ext\_prior}}} \tag{12}$$

where, $\sigma_{\text{ext\_prior}}$ is the two-hour average of $\sigma_{\text{ext}}$ before the NPF start, and $\sigma_{\text{ext\_after}}$ is determined as the average $\sigma_{\text{ext}}$ during the period with NPF influence as described in Sect.2.3.5.





## 3 Results I: Basic features of NPF

### 3.1 NPF frequency

Table 2 presents the frequencies of NPF events on annual basis observed at each site from 2009 to 2013. To eliminate the bias caused by missing data, months with PNSD data coverage less than 75 % were excluded from the frequency statistics. The NPF frequencies at the sites in the same category were found to be similar. Regional background sites had the highest NPF frequency, with an average of about 20 %, followed by urban background sites with an average of about 15 %. NPF events were observed on about 8 % of days at low mountain range sites and only about 3 % of days at the high Alpine site ZSF. A previous study by Nieminen et al. (2018) found similar annual and seasonal frequencies for MEL and HPB. NPF occurred more frequent at the two low mountain range sites than the high Alpine site ZSF. One plausible explanation is that the atmosphere can be influenced by both PBL and FT in high altitude areas (Sun et al., 2021; Herrmann et al., 2015; Rose et al., 2017). NPF was found to be strongly associated with the air parcel vertically transported from lower altitudes, leading to its low NPF occurrence in high altitudes (Bianchi et al., 2016; Shen et al., 2016; Tröstl et al., 2016).

Figure 2 compares the overall NPF frequencies at the GUAN sites with those of other sites in Europe (Baalbaki et al., 2021; Boulon et al., 2011; Bousiotis et al., 2019; Bousiotis et al., 2021b; Brines et al., 2015; Dameto De España et al., 2017; Herrmann et al., 2015; Hofman et al., 2016; Joutsensaari et al., 2018; Lee et al., 2020; Manninen et al., 2010; Németh et al., 2018; Nieminen et al., 2014; Plauskaite et al., 2010; Salma and Németh, 2019; Sellegri et al., 2019; Smejkalova et al., 2021; Vaananen et al., 2013; Vana et al., 2016). For detail information on the locations of those observation sites and study periods, please refer to Table S1 in the supplementary material.

The NPF frequencies at the three urban background sites in GUAN was found to be similar to those of other urban background sites in Central Europe, such as in Amsterdam (AMS), Budapest (BDP), and Vienna (VIE). The annual NPF frequencies at the three regional background sites in GUAN were at the medium range of all regional background sites, as illustrated in Fig.2. The highest NPF frequency was observed at the site Agia Marina Xyliatos (AMX) in Cyprus. The frequency of NPF and undefined event at AMX were 57 % and 8 %, respectively. Generally, the site-to-site differences in NPF frequency are the result of many factors such as locations, meteorological conditions, and anthropogenic and biogenic emissions in the vicinity of the observation sites (Nieminen et al., 2018). For example, higher NPF frequency was observed at site AMX and CBW than MEL. One possible explanation is that both AMX and CBW are more affected by marine air masses (Manninen et al., 2010; Németh et al., 2018), while MEL is affected more by biogenic emissions from the surrounding forested areas (Bousiotis et al., 2021). When comparing the NPF frequency among different studies, the regional representativeness of individual observation site should be fully considered. Additionally, it needs to be careful when comparing those NPF features since the NPF event was visually classified. The subjective preference in the classification process may introduce bias in NPF frequencies. For instance, the frequencies of NPF events at mountain sites in GUAN were much lower than those of other mountain sites. Especially, the frequency of NPF was only 3.3 % at ZSF, while 14.5 % for another high Alpine site Jungfraujoch (JFJ) by Herrmann et al. (2015). In the visually classification process by Dal Maso et al. (2005), the potential



NPF days can be classified as NPF event or undefined event. As stated by Herrmann et al. (2015), the frequency of NPF event and undefined event are 14.5 % and 5.4 % for JFJ, respectively. The corresponding values are 3.3 % and 15.2 % for ZSF, respectively. This large discrepancy in NPF frequency between these two sites may be resulted from the subjective decision when classifying the dataset into NPF and undefined events during the visual classification process.

Figure 3 shows the monthly NPF frequencies of the nine GUAN sites, and the comparison of the seasonal occurrence frequency of NPF between the nine GUAN sites and other European sites is illustrated in Fig.S2 in the supplementary material. For most sites, the highest occurrence of NPF was found during spring and summer, while the lowest in winter, which was consistent with the observations in previous studies (e.g., Nieminen et al., 2018; Salma and Németh, 2019; Boulon et al., 2011). Such a seasonal pattern is highly related to the seasonal variations of solar radiation and biogenic emissions (Manninen et al., 2010). The seasonal variation in NPF frequency also differed among site categories. In early autumn (September and October), NPF events occurred more frequent in regional background than in urban background sites, likely due to the high emission of biogenic VOCs in rural areas in autumn (Salma et al., 2016). Furthermore, the seasonal pattern of NPF events varied among mountain sites, as shown in Fig. 3j-3i. This variability may be a result of the upslope valley winds, which can have different impact on different sites and seasons depending on the altitude and topography of the site (Nieminen et al., 2018).

## 3.2 Growth and formation rate

Figure 4 shows the basic statistics of annual $GR_{nuc}$ and $J_{nuc}$ at the nine GUAN sites. As listed in Table 2, the annual mean $GR_{nuc}$ for particle sizes of 10−25 nm varied from 3.67 to 4.70 nm h$^{-1}$, with surprisingly minor differences between the sites. Previous studies also found that GR varies little among different sites and exhibits only very weak dependency on the low-volatility vapor concentration, particularly in a fixed site (Kulmala et al., 2022a; Kulmala et al., 2023). However, the site-to-site comparison of $J_{nuc}$ implied that stronger anthropogenic influences could lead to a higher $J_{nuc}$, which was consistent with previous studies (e.g. Bousiotis et al., 2021b; Nieminen et al., 2018; Sebastian et al., 2022). Due to relatively fewer anthropogenic emissions, the $J_{nuc}$ at BOS was notably lower than those observed in LTR and LWE.

Figure 5 displays the annual $GR_{nuc}$ measured at GUAN sites and other European sites (Boulon et al., 2011; Bousiotis et al., 2019; Bousiotis et al., 2021b; Herrmann et al., 2015; Joutsensaari et al., 2018; Lee et al., 2020; Manninen et al., 2010; Nieminen et al., 2014; Németh et al., 2018; Plauskaite et al., 2010; Salma and Németh, 2019; Vaananen et al., 2013; Vana et al., 2016), and the corresponding values please refer to Table S2 in the supplementary material. The annual $GR_{nuc}$ for GUAN sites fall within the range of those reported in previous European studies. Caution should be taken that the differences in measurement settings may influence the comparison between sites. For instance, in Fig.5, the higher $GR_{nuc}$ at sites CBW, VVH, PUY, and OPM may be due to the smaller size range (7−20 nm) used for $GR_{nuc}$ calculation. Nonetheless, the difference of $GR_{nuc}$ among the sites in urban background and regional background categories was smaller than the difference of NPF frequencies (Fig.2). It can be observed that the $GR_{nuc}$ were slightly higher at GUAN sites than those at other sites. This tendency was consistent with another recent study reporting that higher $GR_{nuc}$ were observed in Germany compared to other European countries such as Denmark, Spain, and Finland (Bousiotis et al., 2021b).





Figures 6 and 7 present the seasonal GR$_{nuc}$ and $J_{nuc}$ at GUAN sites in this study. The highest GR$_{nuc}$ were observed in summer for most sites, while the lowest in winter. Many previous studies have also reported such seasonal pattern, especially in regional background area, which have been attributed to enhanced biogenic aerosol precursors and stronger solar radiation during summer (Nieminen et al., 2014; Kerminen et al., 2018; Asmi et al., 2011). Both LTR and LWE are located in the city of

Leipzig, and the different seasonal variation in GR$_{nuc}$ may be due to the data coverage at these two sites. The seasonal variations of $J_{nuc}$ were similar with the one of GR$_{nuc}$ in urban background and regional background sites, with the universal maximum in summer observed. However, a different seasonal pattern for the three mountain sites were observed, with the maximum in $J_{nuc}$ being reached in spring. This seasonal behaviour was observed for the site HPB in another previous study by Nieminen et al. (2018) as well. Another exception to the seasonal pattern of GR$_{nuc}$ and $J_{nuc}$ was NEU, which had clear lower GR$_{nuc}$ and $J_{nuc}$ in

summer than in spring, which may have been underestimated due to missing data.

### 3.3 Starting time of NPF events

Figure 8 shows the estimated starting time of each class I event as a function of the days of year. The black solid line indicates the 14-day moving average starting time. Typically, most NPF events started between 8:00 and 12:00 local time at all GUAN sites. Seasonal variations in starting time were evident, with NPF events starting earlier in summer due to earlier sunrise. It is

important to note that our PNSD observations initiate from particle sizes from 5, 10 or 20 nm for different sites (Table 1). Actual nucleation, i.e. at particle diameters ~ 1−3 nm might have taken place before. Consequently, the starting time presented here could potentially be later than the actual occurrence of nucleation.

It is noteworthy that the differences in the starting time of NPF events exist between sites, as shown in Fig.8 and Fig.S4 in the supplementary material. Geographical location and local sources appear to be associated with the starting time of NPF.

The median starting time for each GUAN site ranged from 10:00 to 11:40 local time. However, the three sites located near Leipzig (LTR, LWE, and MEL) had the earliest starting time around 10:00, while for the three sites located in northern Germany (BOS, WAL, and NEU) it was around 10:30. The three mountainous sites (HPB, SCH, and ZSF) had the latest starting time around 11:30. The earliest starting time near Leipzig may be a result of higher concentration of precursors originated from anthropogenic emission in morning rush hour and the latest starting time in mountainous area may be due to

the slower ambient temperature increase in morning.

## 4 Results II: environmental and climate relevance effects

### 4.1 Contribution of NPF on ultrafine particles

NPF events are believed to be a significant source of UFP. In this section, the contribution of NPF on UFP number concentration were qualitatively and quantitively evaluated using two approaches by Salma et al. (2017) and Ma and Birmili

(2015).



### 4.1.1 Nucleation strength factor

Nucleation strength factor (NSF) is a simple metric to qualitatively estimate the relative concentration increment of UFP number concentration on NPF days. As stated in Salma et al. (2017), an NSF of 1 indicates that the relative contribution of NPF events to UFP is negligible, while a value >2 suggests that NPF can be considered as a dominant source of UFP at the site on NPF days. Figure 9 compares the annual median NSF between the nine GUAN sites and those reported in an earlier study by Bousiotis et al. (2021b). The NSF is around 2 for all regional background and mountainous sites, implying that NPF events are the dominant source on NPF days in those environments. NSF is much lower in urban background sites, typically ranging between 1 and 2. Ma and Birmili (2015) reported that aged traffic and other urban sources contributed around 40 % and 30 % to $N_{5-100}$ and $N_{20-100}$ at LTR, respectively. Higher anthropogenic emissions results in higher UFP number concentration and thus lower NSF in urban area. In addition, such high contributions from anthropogenic sources lead to an increased CS, causing more new particles to be scavenged by the more polluted atmosphere, resulting in lower NSF in urban area.

### 4.1.2 Quantitative contribution to UFP

Another approach was further implemented to derive a quantitative average contribution of NPF to UFP. The diurnal cycles of UFP increment on NPF days with high and low solar radiation was estimated, as described in Ma and Birmili (2015). In our study, the UFPs were assumed to originated from "NPF" and "other sources", in which "other sources" encompassed all non-NPF sources such as fresh local traffic, aged traffic, other urban sources, and regional backgrounds.

Figure 10 displays the absolute and relative contributions of NPF to UFP ($N_{10-100}$) at seven GUAN sites, and Fig.11 shows the monthly relative contributions. The analysis did not include BOS and ZSF due to the absence of solar radiation data. As seen in the two figures, the highest contribution of NPF to UFP were found for the regional background sites, with contributions of around 25 % at MEL and NEU, and 15 % at WAL. For the two urban background sites, LTR and LWE, the contributions of NPF were lower, accounting for 11 % and 15 % of $N_{10-100}$, respectively. As discussed in Ma and Birmili (2015), regional background aerosols contribute to UFP equally for urban background (LTR) and nearby regional background (MEL). However, some urban sources such as aged traffic also contribute to UFP in urban background, resulting in a lower relative contribution of NPF to UFP at urban background sites. Due to the low occurrence and low nucleation rate of NPF at the mountain sites (Table 2), the contribution of NPF to UFP was the lowest at HPB and SCH, accounting for 5 % and 9 % of $N_{10-100}$, respectively.

Pronounced seasonal variation of the relative contributions of NPF to UFP were found for all the seven GUAN sites (Fig.11), with higher contribution from May to August and almost no contribution in winter. The contribution of NPF to UFP is determined by many factors such as the frequency, nucleation rate and growth rate of NPF, as well as the concentration of particles from other sources. The contribution of NPF is proportional to the frequency of NPF if keep other factors unchanged. Therefore, the seasonal patterns of the relative contributions of NPF were very similar to the seasonal variation of NPF frequency for each site (Fig.3). The highest relative contributions of NPF to UFP were observed during summer (from May to



August), with the range of 30 % ~ 48 % and 41 % ~ 56 % at urban background and regional background sites, respectively. However, the seasonal distributions of NPF contribution observed at the mountain sites were similar to the one of NPF
frequency in Fig.3, peaking in spring from March to May with the value from 14 % to 23 %.

## 4.2 Contribution of NPF on cloud nuclei condensation (CCN)

NPF is one of the crucial sources of CCN and may affect the regional climate (Williamson et al., 2019). To evaluate the potential contribution of NPF to CCN, we utilized the approach proposed by the previous studies (Kalkavouras et al., 2019; Ren et al., 2021). Table 3 summarizes the relative enhancement of CCN number concentration ($N_{CCN}$ enhancement, denoted
as $E_{Nccn}$) on NPF days in our study and other previous studies conducted in Europe. Our dataset shows a pattern similar to the results from previous studies (Rejano et al., 2021; Kerminen et al., 2012; Dameto et al., 2017; etc.), with higher $E_{Nccn}$ for weaker influence of anthropogenic emissions. However, exceptions were found for sites BOS and NEU, where $N_{CCN}$ enhancement was much higher than the one at the other sites in the same site categories. The seasonal distribution of $E_{Nccn}$ (Figure S5 in the supplementary material) at BOS indicated that the elevated $N_{CCN}$ enhancement may be due to a significantly
higher $N_{CCN}$ enhancement in autumn. And the higher $N_{CCN}$ enhancement at NEU may be attributed to seasonal bias in data availability. The highest $N_{CCN}$ enhancements were observed in the three mountain sites, due to the low background PNC in those area (Kerminen et al., 2012).

When comparing our results with other studies in Table 3, it is important to proceed with caution that the significant variation in $E_{Nccn}$ may result from different observation periods, assumed supersaturation, critical diameter $D_C$, and $N_{CCN}$
estimation methods. However, some consistencies can still be found. For example, $E_{Nccn}$ at the urban background site Vienna is similar to those at the urban background sites in GUAN, and $E_{Nccn}$ at regional background sites in Finland and Sweden are comparable to those for our sites. Other studies have reported $E_{Nccn}$ for site MEL and HPB as well. The results from the present study are consistent with those from a long-term observation study by Ren et al. (2021), while lower than another short-term NPF case study by Wu et al. (2015).

The observed $E_{Nccn}$ in this study revealed a clear relationship between $E_{Nccn}$ and the degree of anthropogenic emission influence in diverse environments. However, it is important to bear in mind that the estimation of $E_{Nccn}$ was based on a constant $D_C$ and may result in overestimation, as stated by Wu et al. (2015). Besides, the $E_{Nccn}$ estimation accounted only for NPF days, not for the entire observation period. That is, the NPF occurrence frequency was not taken into consideration. It needs to be careful when interpreting the $E_{Nccn}$ values, especially for those high $E_{Nccn}$ values in clean area in Table 3. Accounting for the
high $E_{Nccn}$ but low NPF frequency in those clean areas, it cannot conclude that NPFs have a significant impact on the overall CCN budget at those sites.

## 4.3 Impact of NPF on aerosol extinction coefficient

The growth of newly formed particles into large size during NPF may subsequently affect the bulk aerosol optical properties, and further impact the regional aerosol radiative forcing and climate. However, the impact of NPF on aerosol optical properties





was discussed in only few studies. For instance, Shen et al. (2011) analysed the enhancement of aerosol extinction coefficient during the evolution of a NPF event in a regional background site in China. To investigate the contribution of NPF to aerosol extinction coefficient for diverse environments, the ratio of averaged aerosol extinction coefficient at 550 nm ($\sigma_{ext,\,550\,nm}$) between after and before each NPF event, was evaluated in this section, namely $\sigma_{ext,\,550\,nm}$ enhancement. The start and end point for each NPF event were adopted from those for $N_{CCN}$ enhancement evaluation in Sect.4.2. The $\sigma_{ext,\,550\,nm}$ enhancement

and the corresponding statistically significance are listed in Table 4. Statistically insignificant contributions were found for the other sites, especially for polluted urban sites. However, NPF events occurring in areas with low background PNC and low anthropogenic emissions, such as regional background site NEU and the three mountain sites, can significantly enhance $\sigma_{ext,\,550\,nm}$ on NPF event days. These findings underscore the importance of considering the impact of NPF on optical properties when assessing aerosol radiative forcing, especially in remote regions. Besides, similar with the $E_{N_{ccn}}$ estimation, the

enhancement of $\sigma_{ext,550\,nm}$ was only for NPF days. The results in Table 4 cannot represent the NPF enhancement of $\sigma_{ext,550\,nm}$ over the whole study periods.

When discussing the aforementioned environmental and climate relevance effects of NPF (Sect.4), it is important to bear in mind that the obtained contribution is likely to be underestimated. One reason is the potential of missing cases where NPF is relatively weak or interrupted by changed airmasses during measurements. These occurrences, known as "quiet NPFs", have

been found to contribute to the secondary particles in the atmosphere (Kulmala et al. 2022b). Another reason is that it is difficult to follow the growth of newly formed particles longer than a few hours (certainly less than a day) in a single-site measurement, yet the growing particles remain in the ultrafine range 1-3 days (the time it takes for them to reach CCN size). As a result of these considerations, a substantial portion of the ultrafine particles in the troposphere classified as "background" or "other sources" is actually formed by NPF, either via unclear or weak events or 1-3 days upwind from the measurement site,

leading to an underestimation of the relevance effects discussed above.

## 5 Conclusion

As an important source of sub-micrometer particles, atmospheric new particle formation (NPF) can be observed in various environments in the world and may affect cloud and aerosol optical properties. Studies on NPF have been conducted in diverse environments to understand its mechanism and influencing factors. However, most studies provide little more than snapshots

of the NPF process due to their underlying observations being limited in space and time. To obtain statistically relevant evidence on the occurrence and characteristics of NPF across various environments, long-term measurements covering diverse environments are required. Based on a five-year dataset of the German Ultrafine Aerosol Network (GUAN), we investigated the characteristics of NPF for various environments from urban background to high Alpine, including the occurrence of NPF events, particle formation and growth rates, and the impacts of NPF on ultrafine particles (UFPs), cloud condensation nuclei

(CCN) and aerosol radiative forcing. The NPF features were not only internally discussed for GUAN sites, but also compared with the observations in previous studies, aiming to depict a relatively complete picture of NPF in Central Europe.



NPF is a complicated process and can be affected by various factors including local emissions, atmospheric chemical composition, and meteorological conditions. Regional background sites had the highest NPF frequency, with an average value of about 20 %, followed by urban background sites with an average of 15 %. NPF events were observed on 8 % of days at low mountain range sites and only 3 % of days at the high Alpine site ZSF. The site-to-site differences in NPF frequency can be resulted from several factors such as location, meteorological conditions, or anthropogenic and biogenic emissions in the vicinity of the observation sites. For example, the NPF occurred more frequent in regional background than urban background sites, partially owing to higher anthropogenic emissions and condensational sink (CS) in more polluted urban area.

The annual mean growth rate for particle sizes of $10-25$ nm ($GR_{nuc}$) varied from 3.67 to 4.70 nm h$^{-1}$, while the annual formation rate $J_{nuc}$ from 0.43 to 2.89 cm$^{-3}$ s$^{-1}$ in this study. Increased degree of anthropogenic emissions was found to be strongly associated with increased $J_{nuc}$, implying the crucial role of anthropogenic precursors to NPF.

Some differences in the evaluated NPF features are related with meteorological conditions. For instance, the frequency of NPF, $GR_{nuc}$ and $J_{nuc}$ are much lower at mountain sites than regional background sites, which can be attributed to the influence of meteorological conditions such as lower ambient temperature, different mixing state and free tropospheric airmasses. The seasonal variations of NPF frequency, $GR_{nuc}$ and $J_{nuc}$, starting time were also found to be related with meteorological conditions. However, the influence of meteorological factors is not the main objective of this study, and it will be further discussed in a future study.

The impact of NPF on ultrafine particles (UFPs), cloud condensation nuclei (CCN) and radiative forcing were quantitively evaluated and discussed in this study. Over the entire observation periods, the NPF contributed on UFP was about 13 %, 21 %, and 7 % for the urban background, regional background, and low mountain range sites. The enhancement of NPF on CCN number concentration for NPF days were found to be the highest and the most significant in mountain sites. Similarly, the enhancement of NPF on extinction coefficient at 550 nm ($\sigma_{ext,550\,nm}$) for NPF days were 1.35, 1.78, 1.57, and 1.85 for site NEU, HPB, SCH, and ZSF, respectively, while no statistically significant contributions were observed for other sites. These findings underscore the importance of considering the local environments of NPF when assessing its potential impact on regional climate in models. They also emphasize the usefulness of a long-term aerosol measurement network with multiple sites for understanding the variation of NPF features and their influencing factors over a regional scale.

**Data availability**

Datasets for this paper can be accessed at https://ebas-data.nilu.no/Default.aspx (Birmili et al., 2016).



**Author contributions**

AW, NM, WB, MH, and JS designed the research. JS, YY, and NM conducted the data analysis with the help from VK and MK. KaW, MM, TT, HF, BB, LR, CC, ME, RS, KW, FM, MS, OB, and BH conducted the measurements. JS and NM wrote the paper with input from all co-authors.

**Competing interests**

Veli-Matti Kerminen and Markku Kulmala are members of the editorial board of Atmospheric Chemistry and Physics.


**Disclaimer**

Publisher's note: Copernicus Publications remains neutral with regard to jurisdictional claims in published maps and institutional affiliations.

**Acknowledgement**

We acknowledge funding by the German Federal Environment Ministry (BMU) grants F&E 370343200 (German title: Erfassung der Zahl feiner und ultrafeiner Partikel in der Außenluft) from 2008 to 2010, and F&E 371143232 (German title: Trendanalysen gesundheitsgefährdender Fein- und Ultrafeinstaubfraktionen unter Nutzung der im German Ultrafine Aerosol Network (GUAN) ermittelten Immissionsdaten durch Fortführung und Interpretation der Messreihen) from 2012 to 2013.

The authors would like to thank the technical and scientific staff members of the stations included in this study. André

Sonntag and Stephan Nordmann (TROPOS/UBA) contributed to data processing. This work was also accomplished in the frame of the project ACTRIS-2 (Aerosols, Clouds, and Trace gases Research InfraStructure) under the European Union—Research Infrastructure Action in the frame of the H2020 program for "Integrating and opening existing national and regional research infrastructures of European interest" under Grant Agreement N654109 (Horizon 2020). Additionally, we acknowledge the WCCAP (World Calibration Centre for Aerosol Physics) as part of the WMO-GAW program base-funded

by the UBA.

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



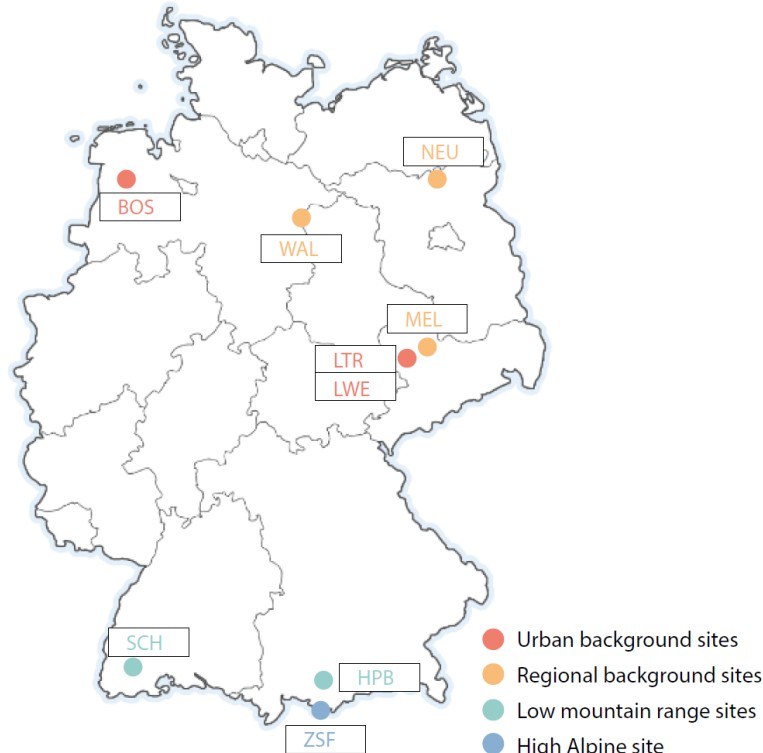

**Figure 1: The locations of the nine selected observation sites in the German Ultrafine Aerosol Network (GUAN).**

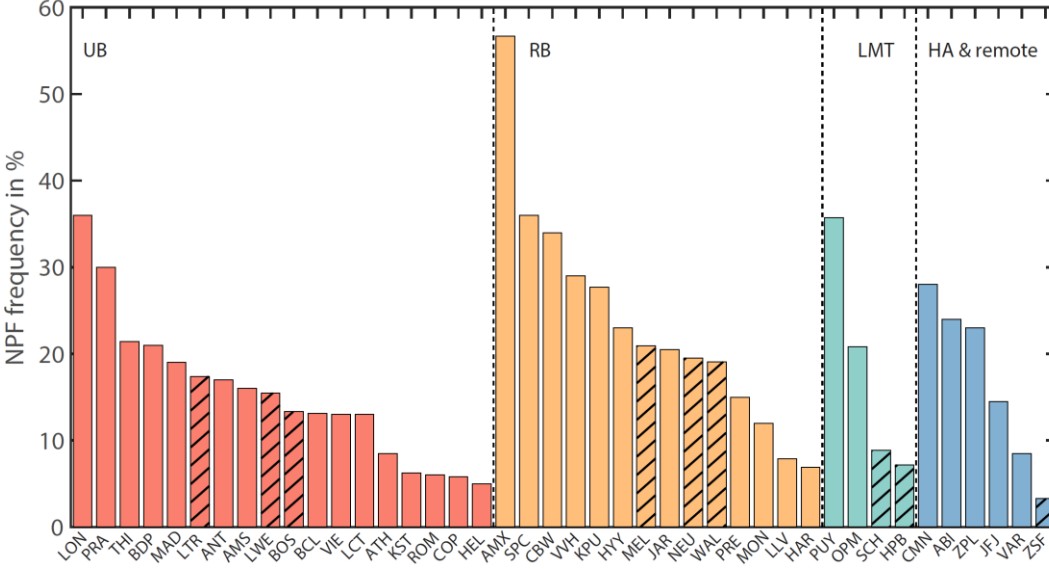

**Figure 2: Annual frequency of NPF events in the present study and other studies in Europe. The hatched pattern denotes the results for the GUAN sites in this study.**



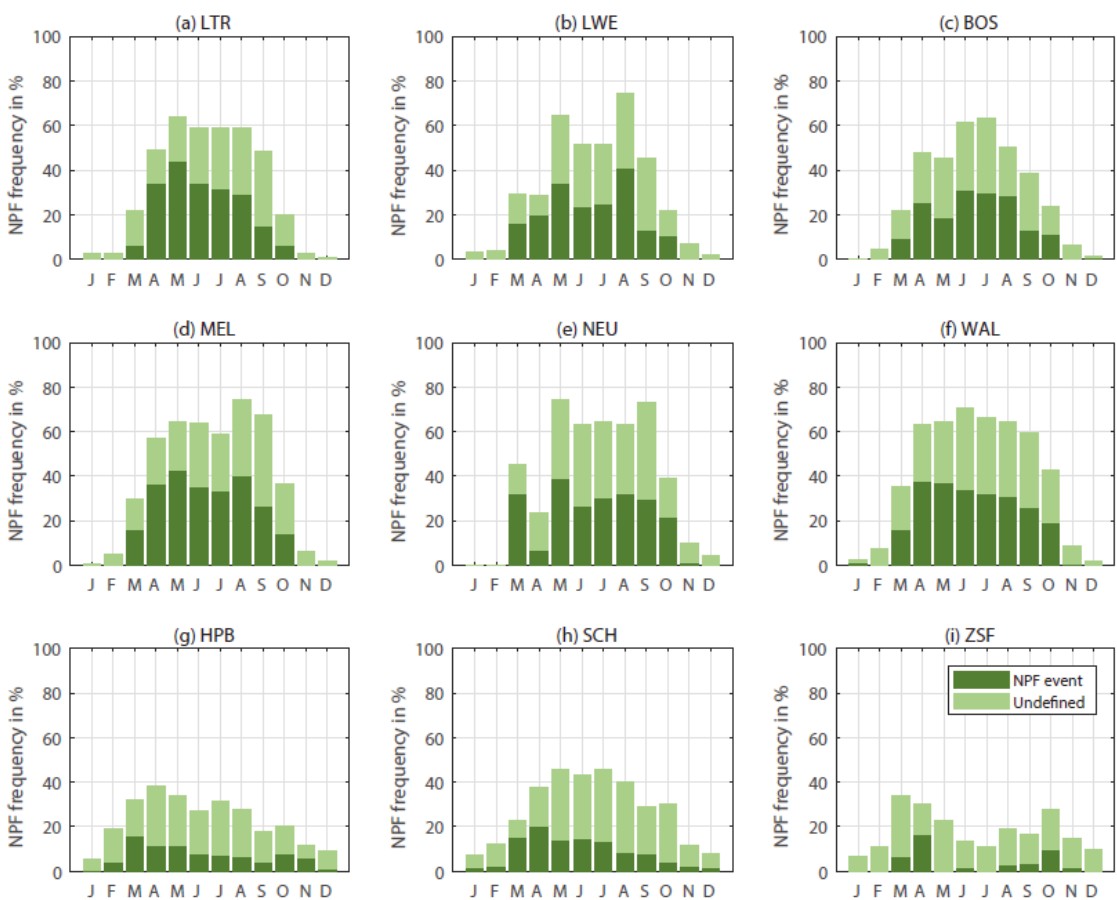

**Figure 3: Monthly frequencies of NPF events for the nine GUAN sites. The dark green bar denotes the frequencies of the NPF event (class I and II), and light green for the undefined events.**

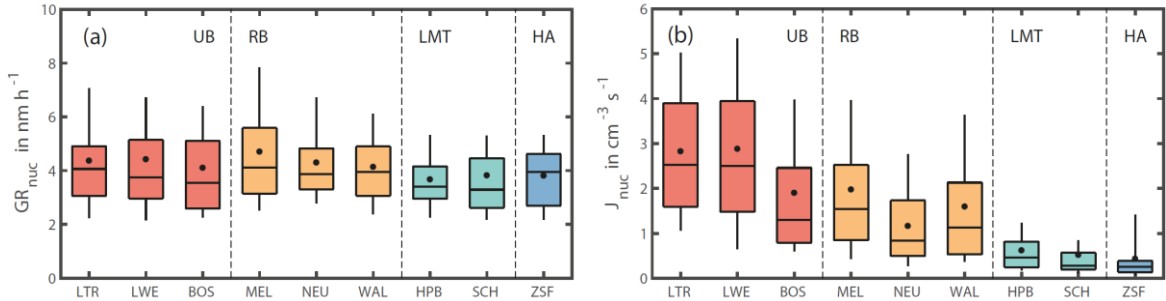


**Figure 4: Basic statistics of GR$_{nuc}$ and $J_{nuc}$ measured at the GUAN sites. Dots denote the mean vales, and the boxes and whiskers denote the 10th, 25th, 50th, 75th, and 90th percentiles.**



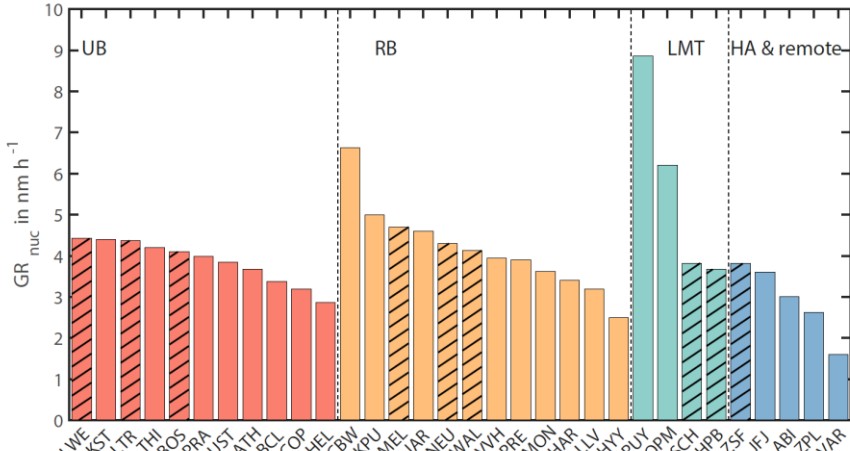

**Figure 5: Average GR$_{nuc}$ in the present study and other studies in Europe. The hatched pattern denotes the results for the GUAN sites in this study.**

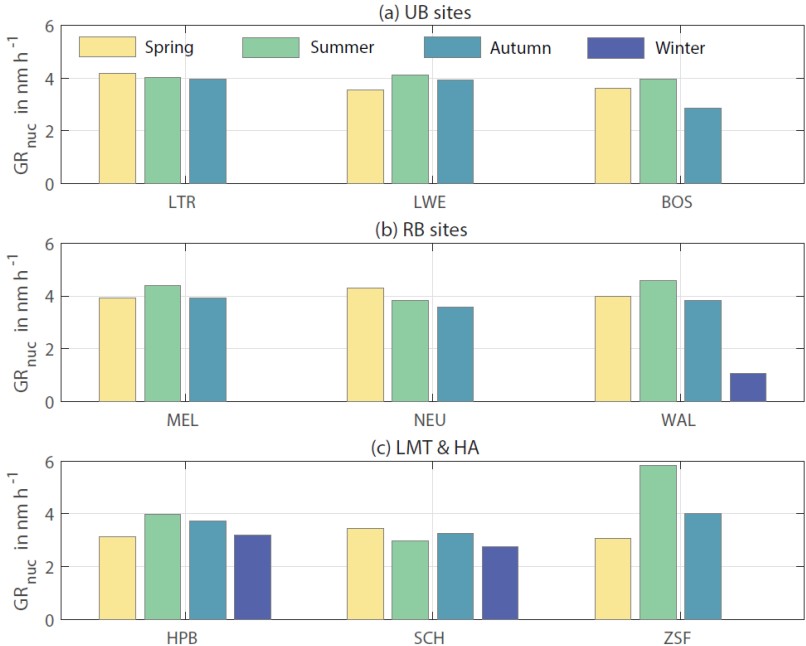

**Figure 6: Seasonal mean GR$_{nuc}$ of NPF events for the nine GUAN sites.**



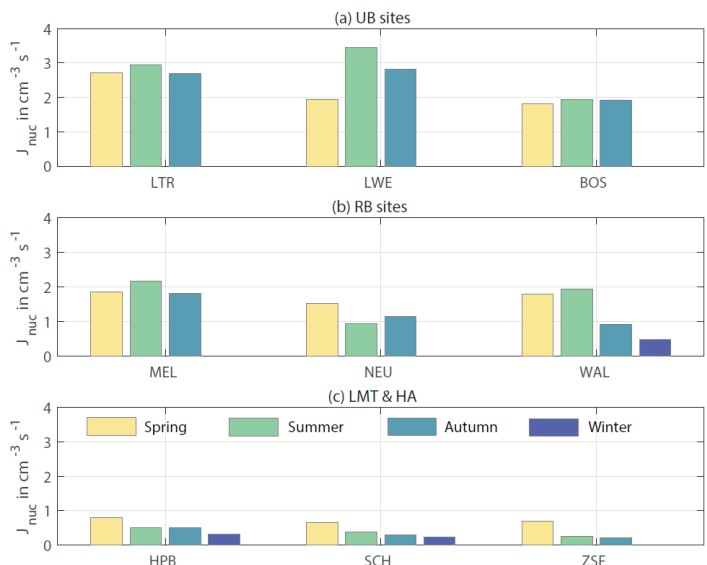

**Figure 7: Seasonal mean $J_{nuc}$ for NPF events at the nine GUAN sites.**

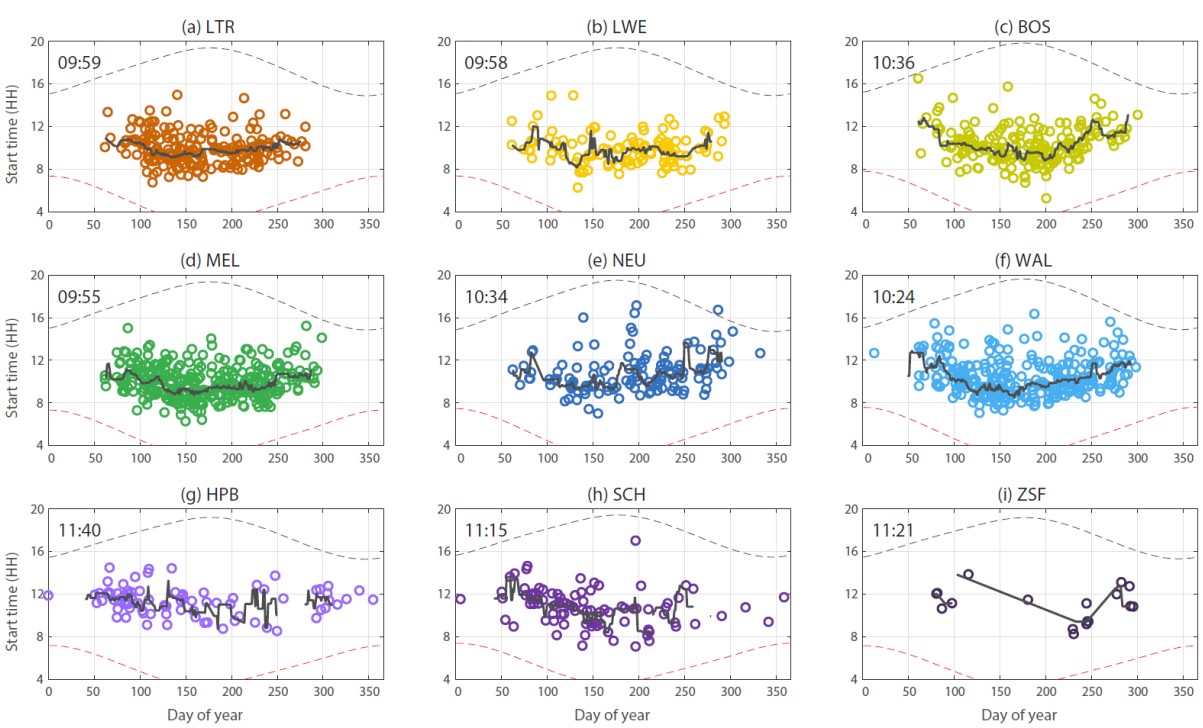


**Figure 8: Scatter plot of NPF starting time depending on days of the year. Black solid lines denote the 14-days moving average of starting time, the red dash and black dash line indicate the sunrise and sunset time, respectively.**



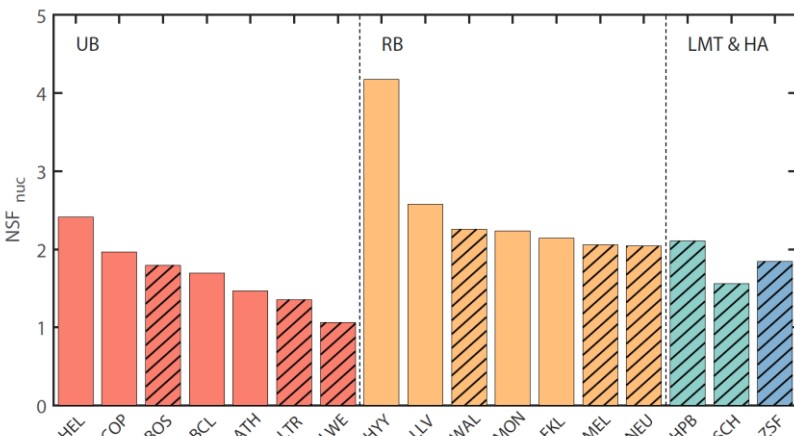

**Figure 9: Median NSF in the present study and other studies in Europe. The hatched pattern denotes the results for the GUAN sites in this study.**

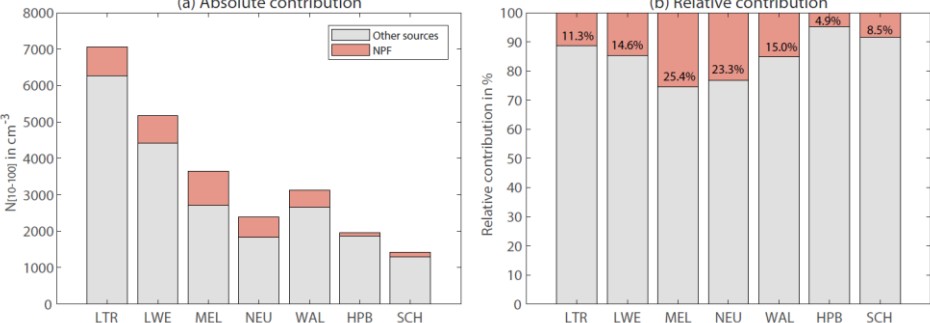

**Figure 10: The absolute and relative contribution of NPF (red) on UFP ($N_{10-100}$) for GUAN sites.**



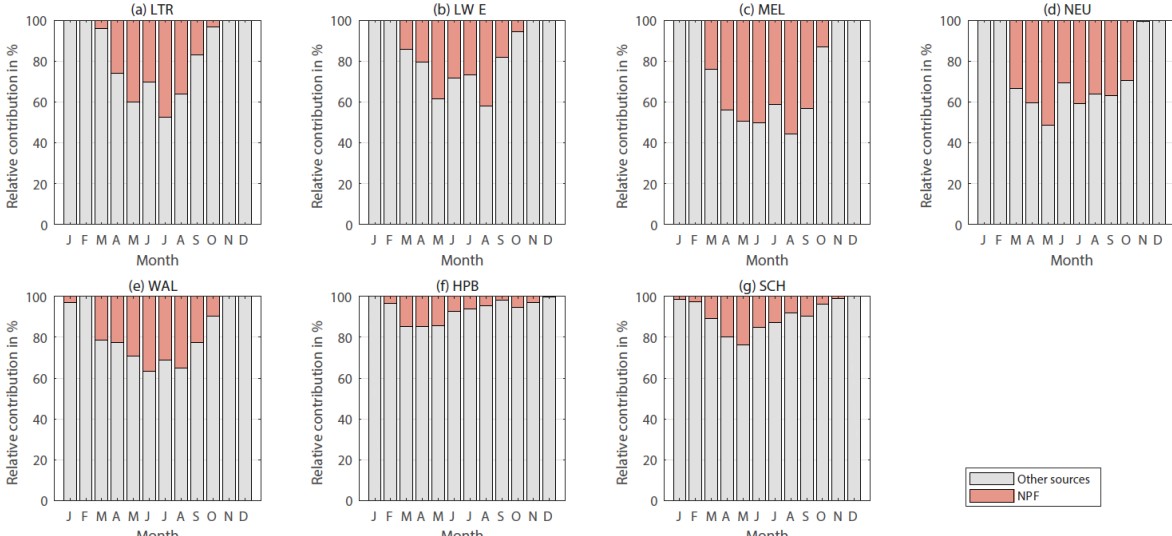

Figure 11: The monthly distribution of relative contribution of NPF on UFP ($N_{10-100}$) in the seven GUAN sites.






**Table 1: Information of the nine GUAN sites and the corresponding PNSD measurements, in alphabetic order.**

| No. | Site name | Abbrevia-tion | Site category | Altitude | Location | MPSS type | Size range |
|-----|-----------|---------------|---------------|----------|----------|-----------|------------|
| 1 | Bösel | BOS | Urban background | 17 m | 52°59'53" N, 07°56'34" E | MPSS | 10–800 nm |
| 2 | Hohenpeißen-berg | HPB | Low mountain range | 980 m | 47°48'06" N, 11°00'34" E | MPSS | 10–800 nm |
| 3 | Leipzig-TROPOS | LTR | Urban background | 126 m | 51°21'10" N, 12°26'03" E | TDMPSS | 5–800 nm |
| 4 | Leipzig-West | LWE | Urban background | 122 m | 51°19'05" N, 12°17'51" E | TDMPSS | 10–800 nm |
| 5 | Melpitz | MEL | Regional background | 86 m | 51°31'32" N, 12°55'40" E | D-MPSS | 5–800 nm |
| 6 | Neuglobsow | NEU | Regional background | 70 m | 53°08'28" N, 13°01'52" E | MPSS | 10–800 nm |
| 7 | Schauinsland | SCH | Low mountain range | 1205 m | 47°54'49" N, 07°54'29" E | MPSS | 10–800 nm |
| 8 | Waldhof | WAL | Regional background | 75 m | 52°48'04" N, 10°45'23" E | MPSS | 10–800 nm |
| 9 | Zugspize Schneeferner-haus | ZSF | High Alpine | 2670 m | 47°25'00" N, 10°58'47" E | MPSS (TSI 3936) | 20–600 nm |






**Table 2: Annual frequency, average growth and formation rate of NPF events at each observation site.**

| Site category | Site name | NPF frequency | Undefined event Frequency | $GR_{nuc}$ in nm h$^{-1}$ | $J_{nuc}$ in cm$^{-3}$ s$^{-1}$ |
|---|---|---|---|---|---|
| **Urban background (UB)** | LTR | 17.4 % | 15.9 % | 4.37 | 2.83 |
| | LWE | 15.5 % | 14.9 % | 4.42 | 2.89 |
| | BOS | 13.3 % | 15.6 % | 4.10 | 1.90 |
| **Regional background (RB)** | MEL | 20.9 % | 17.7 % | 4.70 | 1.98 |
| | NEU | 19.5 % | 22.0 % | 4.30 | 1.16 |
| | WAL | 19.1 % | 20.4 % | 4.13 | 1.59 |
| **Low mountain range (LMT)** | HPB | 7.2 % | 15.9 % | 3.67 | 0.62 |
| | SCH | 8.8 % | 19.4 % | 3.82 | 0.52 |
| **High Alpine (HA)** | ZSF | 3.3 % | 15.2 % | 3.81 | 0.43 |



**Table 3: Comparison of the enhancement of CCN by NPF ($N_{CCN}$ enhancement, $E_{Nccn}$) from multiple European studies.**

| Site | Site category | Time period | CCN method | Critical diameter in nm | Supersaturation (ss) in % | $E_{Nccn}$ | Reference |
|------|------|------|------|------|------|------|------|
| LTR, Germany | UB | 2009−2013 | Calculated | 190, 80, 60 | 0.1, 0.4, 0.6 | 1.01, 1.19, 1.23 | This study |
| LWE, Germany | UB | 2011−2013 | Calculated | 190, 80, 60 | 0.1, 0.4, 0.6 | 0.95, 1.14, 1.16 | |
| BOS, Germany | UB | 2009−2013 | Calculated | 190, 80, 60 | 0.1, 0.4, 0.6 | 1.22, 1.56, 1.61 | |
| MEL, Germany | RB | 2009−2013 | Calculated | 190, 80, 60 | 0.1, 0.4, 0.6 | 1.08, 1.23, 1.34 | |
| NEU, Germany | RB | 2011−2013 | Calculated | 190, 80, 60 | 0.1, 0.4, 0.6 | 1.30, 1.48, 1.51 | |
| WAL, Germany | RB | 2009−2013 | Calculated | 190, 80, 60 | 0.1, 0.4, 0.6 | 1.03, 1.24, 1.27 | |
| HPB, Germany | LMT | 2009−2013 | Calculated | 190, 80, 60 | 0.1, 0.4, 0.6 | 1.61, 1.72, 1.84 | |
| SCH, Germany | LMT | 2009−2013 | Calculated | 190, 80, 60 | 0.1, 0.4, 0.6 | 1.47, 1.77, 1.91 | |
| ZSF, Germany | HA | 2012−2013 | Calculated | 190, 80, 60 | 0.1, 0.4, 0.6 | 1.92, 1.84, 1.87 | |
| Vienna, Austria | UB | 2014−2015 | Measured | 57 | 0.50 | 1.43 | Dameto et al., 2017 |
| University of Crete, Greece | Coastal | 2008−2015 | Measured | 162, 67, 54, 46, 43, 35 | 0.1, 0.38, 0.52, 0.66, 0.73, 1.0 | 1.29−1.77 | Kalkavouras et al., 2019 |
| Sierra Nevada National Park, Spain | High altitude | 2018−2019 | Measured | 66 | 0.5 | 1.75 | Rejano et al., 2021 |
| Hyytiälä, Finland | RB | 2009−2009 | Measured | | 0.1, 0.2, 0.4, 0.8, 1.0 | 2.06, 2.1, 1.7, 1.82, 1.7 | Sihto et al., 2011 |
| MEL, Germany | RB | May − June, 2008 | Calculated | Varied | 0.1, 0.4, 0.6 | 1.63, 1.66.1.69 | Wu et al., 2015 |
| MEL, Germany | RB | May, 2017 | Calculated | | 0.2, 0.4, 0.8 | 0.96, 1.32, 1.72 | Ren et al., 2021 |
| HPB, Germany | LMT | October, 2015 | Calculated | | 0.2, 0.4, 0.8 | 0.85, 1.05, 1.45 | |
| Vavihill, Sweden | RB | October, 2009 | Calculated | | 0.2, 0.4, 0.8 | 0.96, 1.22, 1.51 | |
| RV Polarstern, Norway | Polar | June − July, 2018 | Measured | | 0.1−1.0 | between 2 and 5 | Kecorius et al., 2019 |



**Table 4: The enhancement of extinction coefficient at 550 nm ($\sigma_{ext,550\ nm}$) by NPF for GUAN sites. The bold numbers denote the statistically significant results with $\alpha=0.05$.**

| Site category | Site | $\sigma_{ext,550\ nm}$ enhancement |
|---|---|---|
| UB | LTR | 1.02 |
| | LWE | 0.95 |
| | BOS | 1.22 |
| RB | MEL | 1.06 |
| | NEU | **1.35** |
| | WAL | 1.01 |
| LMT | HPB | **1.78** |
| | SCH | **1.57** |
| HA | ZSF | **1.85** |
