# Peer review of "Measurement report: Contribution of atmospheric new particle formation to ultrafine particle concentration, cloud condensation nuclei and radiative forcing: Results from five-year observations in Central Europe"

_EGUsphere, 2023_

## Author Comment (AC1)

**Response to Reviewer #1:**

**General comments:**

The MS reports, analyses and discusses data on the NPF occurrence frequency, particle formation rate, growth rate, starting time, contribution of NPF events to UF particle number concentration, to CCN concentration and to aerosol extinction coefficient obtained within the German Ultrafine Aerosol Network for 5 years. The topic of the MS is timely and of interest for the international scientific community. The experimental and evaluation methods were carefully deployed and realised. The obtained results are very valuable considering both the spatial and temporal scales. They are also put into an international context by comparing them with those from some other, mainly central European sites. Nevertheless, there are several limitations which should be corrected and some aspects in which the MS could be and should be extended.

Response:

We appreciated reviewer's positive feedback and constructive suggestions which are of great value for improving the quality of our paper. Our point-to-point replies to the reviewer's comments are listed below.

**Major concerns**

1. The formation rate and growth rate were calculated for the diameter of 10 nm (Eqs. 1 and 2). Nevertheless, they are denoted as Jnuc and GRnuc. This is misleading since the subscript nuc usually indicates the properties at the nucleation, i.e. at a diameter of ca. 1.5 nm. Since the dynamic properties can strongly change with particle diameter in this range, this practice is not acceptable. The authors should use J10 and GR10 instead of the present notation all over the MS. More importantly and a consequence of this, it should be considered and discussed in more detail that the comparisons of these data to the other European results were accomplished at different diameters, e.g. to 3 or 6 nm. It is also noted in this respect that 1) the mean diameter (L144) should be replaced by the modal diameter, and 2) it should be specified for which particle diameter Dp was the CoagSDp calculated.

Response:

Thanks for the comments. We agree with the reviewer and have revised the text as below:

(1) We have replaced $J_{nuc}$ and $GR_{nuc}$ with $J_{10\text{-}25}$ and $GR_{10\text{-}25}$, respectively all over the manuscript.

(2) In the description of calculation of growth and formation rates in Sect.2.3.2, the "mean diameter" has been replaced as "modal diameter" and the size of $CoagS_{Dp}$ was clarified as 10 nm.

(3) We have clarified the individual size range of GR and $J$ in other European studies in Table S2 in the supplement material.

(4) We have extended the discussion on the comparison of GR and $J$ between GUAN and other European sites in Sect.3.2. The revised paragraphs are as follows:

- Revision in Sect.2.3.2:

  "The growth and formation rate were evaluated for class I event in this study, while CS for all NPF events. The growth rate of nucleation mode particles ($GR_{10\text{-}25}$) is defined as the change rate of the modal diameter of the newly formed particles (Kulmala et al., 2012)…"

  "$J_{10-25} = \frac{dN_{10-25}}{dt} + CoagS_{10nm} \times N_{10-25} + \frac{GR_{10-25}}{\Delta D_p} \times N_{10-25}$ \hfill (2)

  where $CoagS_{10nm}$ is the coagulation sink of particles with diameter of 10 nm, which can be calculated using the method proposed by Kerminen et al. (2001):

  $CoagS_{10nm} = \sum_{D_p'=10nm}^{D_p'=800nm} K(10nm, D_p') N_{D_p'}$ \hfill (3)

  where $K(10nm, D_p')$ is the coagulation coefficient between particles with sizes of 10 nm and $D_p'$, and $N_{D_p'}$ is the particle number concentration of particle with size $D_p'$."

- Extended discussions in Sect.3.2:

  "Figure 5 displays the annual GR measured at GUAN sites and other European sites (Boulon et al., 2011; Bousiotis et al., 2019; Bousiotis et al., 2021b; Herrmann et al., 2015; Kalkavouras et al., 2020; Lee et al., 2020; Manninen et al., 2010; Nieminen et al., 2014; Nieminen et al., 2018; Salma et al., 2016; Tröstl et al., 2015; Vaananen et al., 2013; Vana et al., 2016). For the GR values and the corresponding size range reported in those studies please refer to Table S2 in the supplementary material. The $GR_{10\text{-}25}$ for GUAN sites falls within the range of those reported in previous European studies. Caution should be taken that the differences in observation periods and size ranges of GR may influence the comparison among sites. In UB sites, the highest GR was reported at BUD, with the size range of 6–50 nm. LWE, KST and LTR showed the similar GR level, but the size range of GR at KST was 16.6–50 nm. The lowest GR in UB sites were observed at COP and HEL, with the evaluated size range of 5.8–30 and 3.4–30 nm, respectively. In RB sites, the GR at site CBW was about 6.6 nm h$^{-1}$, which was much higher than other RB sites. This high GR at CBW may be

resulted from the short observation period in this study, from 1 Apr 2008 to 31 Mar 2009 (Manninen et al., 2010). Meanwhile, another study reported the seasonal variation of GR at CBW between 10 and 25 nm as well (Nieminnen et al., 2018). The seasonal $GR_{10-25}$ ranged from 2.9 to 4.9 nm h$^{-1}$, which was similar with $GR_{10-25}$ at RB sites of GUAN. For LMT sites, the $GR_{10-25}$ at SCH and HPB were lower than the $GR_{7-20}$ at another two LMT sites PUY and OPM located in central France. Nieminen et al. (2018) also found that $GR_{10-25}$ at PUY was significantly higher than those at other LMT sites, possibly related to the vertical transport of particles within the boundary layer. For high altitude and remote sites, the $GR_{10-25}$ of ZSF was comparable to those of other sites."

2. The authors mention (L290) that the starting times (t1) of the NPF event were determined at different particle sizes of 5, 10 or 20 nm (lower diameter limit of the measurement setups), which makes the discussion and comparison difficult. This particle diameter variability could be taken into consideration in the first approximation when the starting times are shifted to the critical nucleation diameter of ca. 2 nm using the GR values which are the closest to the actual particle diameter by a subtraction of e.g. t1-(10-2)/GR10 (for 10 nm). This modified starting time tnuc could be compared more advantageously despite our knowledge on the GR in the diameter range from 2 to 3 or 5 nm is still inconclusive.

Response:

Thanks for the suggestion. We have converted the starting time from 10 nm ($t_{10\,nm}$) to 2 nm ($t_{2\,nm}$) according to the reviewer's suggestion. The hypothesis behind this estimation is that the $GR_{10-25}$ equals to $GR_{2-10}$. To verify the effectiveness of this conversion, we have compared $GR_{10-25}$ and $GR_{5-25}$ for MEL in the year of 2012 since the PNSD have been measured down to 5 nm. The relative difference between $GR_{5-25}$ and $GR_{10-25}$ is only 1 %, meaning only negligible bias will be introduced in the conversion.

The scatter plot of starting time $t_{2\,nm}$ is shown in Fig.R1. The $t_{2\,nm}$ was about 2 hours earlier than $t_{10\,nm}$. We have updated Fig.8 and corresponding text in Sect.3.3:

"Figure 8 shows the estimated starting time of class I events as a function of day of year. The starting time was initially estimated based on local time (UTC+1) and further converted to solar time according to the longitude of the sites. The PNSD observations in our dataset initiate from particle sizes from 5 or 10 nm at different sites (Table 1). However, starting time at 10 nm ($t_{10\,nm}$) was not able to describe the actual occurrence time of nucleation. Therefore, $t_{10\,nm}$ has been converted to the critical nucleation diameter of 2 nm ($t_{2\,nm}$) using the $GR_{10-25}$ values by $t_{2\,nm} = t_{10\,nm} - \frac{(10-2)}{GR_{10-25}}$.

Typically, most NPF events started between 07:30 and 9:00 solar time at all GUAN sites. Seasonal variations in starting time were evident, with earlier starting time in summer due to earlier sunrise. It is noteworthy that the differences in the starting time of NPF events exist between sites, as shown in Fig.8 and Fig.S7 in the supplementary material. The three mountainous sites (HPB, SCH, and ZSF) had the latest starting time around 09:00. Two UB sites LTR and LWE had the earliest starting time around 07:30. Starting time at BOS and three RB sites (MEL, WAL, and NEU) is around 08:30. Since the use of solar time has already eliminated the bias of local time relative to site longitude, the difference of starting time among sites mainly stems from the different diurnal variation of precursor concentration and CS, especially in urban background area."

[Figure]

**Figure R1: Scatter plot of NPF starting time at 2 nm depending on days of the year. The time displayed in the figure is the overall mean starting time. Black solid lines denote the mean value of starting time at each season, and the red and black dash line indicate the sunrise and sunset time, respectively.**

3. Local time was used in the work as the time base (L136, L288). This selection (as all the other options as well) has both advantages and limitations. It should be clarified weather the starting times in Fig.8 were adjusted to clock change (expressed in UTC+1) or not. In relation to this, the reader may also wonder what the reasons were for selecting the 14-day smoothing in Fig.8 and not something else.

Response:

To better illustrate the influence of local environment on the NPF characteristics, the starting time evaluated in this study was the solar time, which was adjusted from local time

(UTC+1) according to the longitude of the measurement sites. We have emphasized this point in the text of Sect.3.3. Figure 8 has been revised accordingly.

"Figure 8 shows the estimated starting time of class I events as a function of day of year. The starting time was initially estimated based on local time (UTC+1) and further converted to solar time according to the longitude of the sites...

…Since the use of solar time has already eliminated the bias of local time relative to site longitude, the difference of starting time among sites mainly stems from the different diurnal variation of precursor concentration and CS…."

The reason for selecting the 14-day smoothing in Fig.8 is to show the seasonal variation of starting time. To make it clearer, we have replaced the 14-days smoothing to seasonal mean starting time, illustrated as black solid line in Fig.R1.

4. The authors are requested to explain why they can show occurrence frequency data for the winter months (e.g. Fig.3), while the column bars of nuc and GRnuc for winter are mostly missing in Figs. 6 and 7. Cf. lines 212-213.

Response:

Thanks for the comment. The GR and $J$ were only estimated for class I NPF events. As shown in Fig.3, NPF events were observed in winter at sites NEU, WAL, HPB, and SCH. But the winter-time NPF events at NEU were all class II events, thus no winter-time growth and formation rates were estimated for NEU in Fig.6 and 7.

Accordingly, we have clarified this point in "Sect.2.3.2" and "Sect.3.2":

● Revision in Sect.2.3.2:

"The growth rate and formation were evaluated for class I event in this study, while CS for all NPF events."

● Revision in Sect.3.2:

"Figure 4 shows the basic statistics of annual $GR_{10-25}$, $J_{10-25}$ and CS at the nine GUAN sites. The growth and formation rate were only evaluated for class I event in this study, and the CS was estimated for all NPF event days…"

"Figures 6 and 7 present the seasonal $GR_{10-25}$ and $J_{10-25}$ at GUAN sites in this study. Since $GR_{10-25}$ and $J_{10-25}$ were only evaluated for class I events, there were NPF events observed in winter-time but no $GR_{10-25}$ and $J_{10-25}$ evaluated at some sites, for example at NEU."

5. The statement in L298-L299 is only partially acceptable. The occurrence and timing of the NPF events depend more sensitively on the ratio of the sources and sinks than on the

sources (precursor emissions) alone. The authors may want to add new aspects on the effects of the higher CS in cities.

Response:

Thanks for the suggestion. Following your advice, we have plotted the diurnal cycle of condensation sink (CS) on NPF days for all the nine GUAN sites in Fig.R2. Since the use of solar time has already eliminated the bias of local time relative to site longitude, the difference of starting time among sites mainly stems from the different diurnal variation of precursor concentration and CS. Unfortunately, we do not have measurements of precursors at the sites. It can be seen in Fig.R2 that the CS at UB sites increases rapidly during morning rush hour due to the strong traffic emission in urban area, implying also an increase of precursor concentration. The ratio of sources and sinks may be changed during this time period and further leads to earlier NPF starting time in urban area than RB sites. In the mountain area, the CS starts to increase at about 08:00 and reaches its daily maximum in the late afternoon, meaning that it may take some time for the development of the boundary layer and transport of the precursors upward after sunrise, resulting in late NPF starting time.

[Figure]

**Figure R2: Mean diurnal cycle of condensation sink during NPF days at GUAN sites.**

Accordingly, the text in "Sect.3.3 Starting time of NPF events" has been revised and Fig.R2 has been added as Fig.S8 in the supplementary material.

[revised manuscript text omitted]

2. The authors may want to revisit their rounding off strategy at many places in the text and tables (e.g. 2.89 cm$^{-3}$ s$^{-1}$ in L25, Table 2), since the anticipated precision of these values seem exaggerated.

Response:

   Thanks for the comment. We have checked and revised all the anticipated precision of numbers in the tables and text.

3. **L31-L32**. It is unusual to state that the particles are formed from precursors. Those chemical compounds (usually with longer atmospheric residence times) which yield the active players (usually with shorter residence times) in reactions are ordinarily called precursors. Thus, SO2 is a precursor compound, whereas its gas-phase oxidation product of H2SO4 is the vapour that plays an active role in the nucleation process. Clarification is needed.

Response:

We agree with the reviewer and revised the text as:

"Atmospheric new particle formation (NPF) is a process initiated with the sudden formation of new particles with diameters less than 3 nm in the atmosphere. Low volatile gas molecules oxidated from gas-phase precursors cluster together and form new aerosol particles. These nano particles may subsequently grow into larger sizes by condensation or coagulation (Kulmala et al., 2014)."

4. **L32-L33.** The sentence is misleading. Condensation does not increase the particle number concentrations, while coagulation decreases them. These processes do not lead to growing particle number concentrations. Reformulation is required.

Response:

    Thanks for the comment. This sentences have been revised as: "Atmospheric new particle formation (NPF) is a process initiated with the sudden formation of new particles with diameters less than 3 nm in the atmosphere. Low volatile gas molecules oxidated from gas-phase precursors cluster together and form new aerosol particles. These nano particles may subsequently grow into larger sizes by condensation or coagulation (Kulmala et al., 2014). The newly formed aerosol particles have the potential to contribute greatly to the number concentration of ultrafine particles (UFPs, particles smaller than 100 nm) or even larger sub-micrometer particles (particles smaller than 1 μm) (Ma and Birmili, 2015)".

5.  **L90 and L125.** Check the citing format requirements of the journal.

Response:

    The format of all the references have been checked and revised according to the requirement of ACP.

6.  **L114.** Replace "Aerosol PNSD was measured" by "Aerosol PNSDs were measured", and similarly: "PNSD were generally measured" by "PNSDs were generally measured" (L118).

Response:

    We have revised the expression of singular and plural words in the whole manuscript.

7.  L158. The NSF was actually introduced and improve in Salma at al., 2017 and not in the reference cited.

Response:

    Thanks. This article has been added in the reference list:

    Salma, I., Varga, V., and Németh, Z.: Quantification of an atmospheric nucleation and growth process as a single source of aerosol particles in a city, Atmos. Chem. Phys., 17, 15007-15017, doi: 10.5194/acp-17-15007-2017, 2017.

8.  The authors may want to write in the section title of 3.1 NPF "occurrence" frequency, and of 3.2 Growth and formation rate"s".

Response:

    Thanks, and the titles of Sect.3.1 and 3.2 have been revised following the reviewer's suggestion.

9. **L243-L245**, twice. The order of the words and grammar should be checked: "the frequency of NPF event and undefined event are 14.5 % and 5.4 % for JFJ, respectively". Consider: the frequencies of the NPF events and undefined events were 14.5 % and 5.4 %, respectively for JFJ.

Response:

Thanks, and the sentences have been revised as:

"As stated by Herrmann et al. (2015), the frequencies of NPF event and undefined event are 14.5 % and 5.4 %, respectively for JFJ. The corresponding values are 3.3 % and 15.2 %, respectively for ZSF."

10. **L261**. The correct formulation is (Kulmala et al., 2022a, 2023).

Response:

Thanks, and the sentence has been revised as:

"Previous studies also found that GR varies little among different sites and exhibits only very weak dependency on the low-volatility vapor concentration, particularly in a fixed site (Kulmala et al., 2022a, 2023)".

11. **L342-L345**. Remove the considerable repetition.

Response:

Thanks. The sentence has been revised as:

"To evaluate the potential contribution of NPF to CCN, the relative enhancement of CCN number concentration ($N_{\text{CCN}}$ enhancement, denoted as $E_{N\text{ccn}}$), which is the ratio between $N_{\text{CCN}}$ after and prior to the NPF event, has been estimated following the approach proposed by previous studies (Kalkavouras et al., 2019; Ren et al., 2021)."

12. Caption of Fig.8. Correct: "…the red dash and black dash line indicate…".

Response:

Thanks. The caption of Fig.8 has been revised as:

"Figure 8: Scatter plot of NPF starting time (solar time) depending on days of the year. Black solid lines denote the mean seasonal starting time, the red and black dash line indicate the sunrise and sunset time, respectively."

---

## Author Comment (AC2)

**Response to Reviewer #2:**

The authors analyse data collected over a five year period at the nine sites of the German Ultrafine Aerosol Network (GUAN) in relation to new particle formation (NPF) events. They compare the behaviour between different site types in terms of important variables such as nucleation frequency, nucleation rate, particle growth rates and implications for incremental ultrafine particles and CCN. They seek to place the results in the context of other European sites. The research is carefully conducted with methods clearly described, and the results, coming from a high quality harmonised network, are of value to the research community.

Response:

We appreciated reviewer's positive feedback and constructive suggestions which are of great value for improving the quality of our paper. Our point-to-point replies to the reviewer's comments are listed below.

1. Line 213. Exclusion of months with data recovery <75% could bias the dataset more than leaving the data in or extrapolating to the full month, as if it excludes a partial month with high or low values, exclusion will cause bias. There is also a question over the fact that site data collected over different periods (of years) are being compared without considering whether the shorter time periods are representative in relation to the longer datasets. The issue of inter-annual variability is not explored in depth, and perhaps should be.

   Response:

   Thanks for the constructive comments.

   (1) Bias in NPF occurrence frequency caused by data exclusion

   To estimate the bias related to data exclusion, we reevaluated the annual frequency of NPF based on the whole dataset. Table R1 lists the results for the whole dataset and the dataset excluding the months with data coverage lower than 75%. We can see that the NPF occurrence frequencies based on the whole dataset are slightly lower than those based on the dataset with data exclusion for most stations. The highest bias was found at SCH at which the occurrence frequencies changed from 8.8 % to 7.8 % for NPF events and from 19.4 % to 17.3 % for undefined events, respectively. This bias may be caused by the lower data coverage at SCH in cold season (September to March). To eliminate this bias, we decided to follow the reviewer's suggestion, using the whole dataset in the revised manuscript.

**Table R1: Annual frequency of NPF events at each observation site, calculated based on the dataset above 75% monthly data coverage and the whole dataset, respectively.**

| site category | site | exclude months with data coverage < 75% | | whole dataset | |
|---|---|---|---|---|---|
| | | NPF event | undefined event | NPF event | undefined event |
| urban background (UB) | LTR | 17.4 % | 15.9 % | 16.8 % | 15.7 % |
| | LWE | 15.5 % | 14.9 % | 15.5 % | 14.5 % |
| | BOS | 13.3 % | 15.6 % | 12.6 % | 14.6 % |
| regional background (RB) | MEL | 20.9 % | 17.7 % | 19.6 % | 16.9 % |
| | NEU | 19.5 % | 22.0 % | 18.3 % | 20.6 % |
| | WAL | 19.1 % | 20.4 % | 19.0 % | 19.9 % |
| low mountain range (LMT) | HPB | 7.2 % | 15.9 % | 6.8 % | 15.4 % |
| | SCH | 8.8 % | 19.4 % | 7.8 % | 17.3 % |
| high Alpine (HA) | ZSF | 3.3 % | 15.2 % | 3.4 % | 14.3 % |

Accordingly, the corresponding text, figures, and table in Sect.3.1 have been revised as follows.

"The NPF occurrence frequencies at the sites in the same category were found to be similar. The regional background sites had the highest NPF occurrence frequency, with an average of about 19 %, followed by the urban background sites with an average of about 15 %. NPF events were observed on about 7 % of days at the low mountain range sites and only about 3 % of days at the high Alpine site ZSF."

[Figure]

**Figure 2: Annual occurrence frequency of NPF events in the present study and other studies in Europe. The hatched pattern denotes the results for the GUAN sites in this study.**

[Figure]

**Figure 3: Monthly occurrence frequencies of NPF events for the nine GUAN sites. The dark green bar denotes the occurrence frequencies of the NPF event (class I and II), and light green for the undefined events.**

**Table 2: Annual occurrence frequency, average growth and formation rates of NPF events at each observation site.**

| Site category | Site name | NPF occurrence frequency | Undefined event occurrence frequency | $GR_{10\text{-}25}$ in nm h$^{-1}$ | $J_{10\text{-}25}$ in cm$^{-3}$ s$^{-1}$ |
|---|---|---|---|---|---|
| Urban background | LTR | 16.8 % | 15.7 % | 4.4 | 2.8 |
| (UB) | LWE | 15.5 % | 14.5 % | 4.4 | 2.9 |
| | BOS | 12.6 % | 14.6 % | 4.1 | 1.9 |
| Regional background | MEL | 19.6 % | 16.9 % | 4.7 | 2.0 |
| (RB) | NEU | 18.3 % | 20.6 % | 4.3 | 1.2 |
| | WAL | 19.0 % | 19.9 % | 4.1 | 1.6 |
| Low mountain range | HPB | 6.8 % | 15.4 % | 3.7 | 0.6 |
| (LMT) | SCH | 7.8 % | 17.3 % | 3.8 | 0.5 |
| High Alpine (HA) | ZSF | 3.4 % | 14.3 % | 3.8 | 0.4 |

(2) The short-term data representativeness:

We fully agree that the inter-annual variability of NPF event might influence the comparison among sites since the data coverages at the nine sites vary from two to five years, especially for the site ZSF with only two-year data available. To evaluate this potential influence, we have firstly calculated the seasonal NPF occurrence frequency for each year at all the nine GUAN sites, as shown in Fig.R1 below.

[Figure]

**Figure R1: Seasonal NPF occurrence frequency at the nine GUAN sites. The values in each year are, in turn, the frequencies of spring (MAM), summer (JJA), autumn (SON) and winter (DJF). The gray dotted line denotes the annual mean occurrence frequency.**

Among the three UB sites, BOS had relatively complete data coverage, while LTR and LWE had missing data in 2013 and 2009−2010, respectively. The sites LTR and LWE are both located in the city of Leipzig and are only 10 km apart. So that the NPF occurrence frequencies at the two sites were very close during the overlapped period from June 2011 to 2012 in Fig.R1. Meanwhile, no significant inter-annual variation was found in the four-year dataset of LTR and three-year dataset of LWE. For RB sites, the observation at NEU was unavailable from 2019 to 2010. Both NEU and WAL can represent the regional background air in the northern Germany lowlands. And no significant inter-annual variation in NPF occurrence frequency was found at WAL.

Hence, we assume that the influence of the inter-annual changes on the characteristics of NPF in LTR, LWE and NEU are limited and the available dataset of these three sites can represent the overall characteristics of NPF for the five-year period.

Among the three mountainous sites, ZSF had the least valid data, with only 2012 and 2013 available. Previous studies have found that the NPF at high altitude sites is mainly affected by the vertical transport of airmasses (Bianchi et al., 2016). In our previous study (Sun et al., 2021), the inter-annual variations of regional and long-range transport airmass were estimated for the sites ZSF and Jungfraujoch (JFJ) from 2009 to 2018. As illustrated in Fig.R2 (Fig.8 in Sun et al. (2021)), the regional airmass occurrence frequency at ZSF increased slightly with a rate of 0.96 %/year from 2009 to 2013. More frequent regional airmass may result in more frequent vertical transport of precursor gases to high-altitudes, leading to higher probability of NPF. The occurrence of NPF depends on several local conditions such as precursors concentration, condensation sink, temperature and solar radiation, etc. However, the inter-annual variation of regional airmass frequency may imply that the characteristics of NPF at ZSF for 2012 and 2013 might be slightly biased with those for the whole period 2009−2013.

[Figure]

**Figure R2: The annual relative occurrence frequency of the two airmasses clusters at two high Alpine sites ZSF (a) and Jungfraujoch (JFJ, b). The figure was published as Fig.8 in Sun et al. (2021).**

To better clarify the influence of data coverage on its representativeness, detailed discussion has been added in "Sect.3.1 NPF occurrence frequency", "Sect.5 Conclusion" and supplementary material:

● Text in Sect.3.1 NPF occurrence frequency:

"It should be noted that there are data missing for one to three years in four of the nine GUAN sites (Fig.S1). A question raised is whether the data missing may cause an

issue of data representativeness. For UB sites, LTR and LWE had missing data in 2013 and 2009−2010, respectively. The sites LTR and LWE are both located in the city of Leipzig and are only 10 km apart. As can be seen from Fig.S2, the NPF occurrence frequencies at the two sites were quite close during the overlapped period from June 2011 to 2012. Meanwhile, no significant inter-annual variation was found in the four-year data of LTR and three-year data of LWE. For RB sites, the observation at NEU was unavailable from 2019 to 2010. Both NEU and WAL can represent the regional background air in the northern Germany lowlands. And no significant inter-annual variation in NPF occurrence frequency was found at WAL. Hence, we assume that the influence of the inter-annual changes on the characteristics of NPF in LTR, LWE and NEU are limited and the available dataset of these three sites can represent the overall characteristics of NPF for the five-year period. Among the three mountainous sites, ZSF had the least valid data, with only 2012 and 2013 available. The regional airmass occurrence frequency at ZSF increased slightly with a rate of 0.96%/year from 2009 to 2013 (Sun et al., 2021), resulting in more frequent vertical transport of precursor gases to high-altitudes. The occurrence of NPF depends on several local conditions such as precursors concentration, condensation sink, temperature and solar radiation, etc. However, the inter-annual variation of regional airmass frequency may imply that the characteristics of NPF at ZSF for 2012 and 2013 might be slightly biased with those for the whole period 2009−2013."

**2.** The explanations for lower nucleation frequencies at high altitude sites are not very convincing. Temperature is mentioned as a contributor to this effect, which is unlikely as lower temperatures serve to stabilise the clusters which are the precursors of new particles. Given this fact and the likely lower condensation sink, the most likely explanation would appear to relate to gaseous precursor concentrations.

Response:

Thanks for the comment. We agree with the reviewer that given the low condensation sink at ZSF (Fig. R3), the low concentration of precursors is likely to be the main reason of the low NPF occurrence frequency. Unfortunately, measurements of gaseous precursors were not available for the period 2009−2013 at ZSF. As illustrated by an earlier study (Flentje et al., 2010), the median $SO_2$ concentration during the year 2000−2007 at ZSF was about 0.18 $\mu g/m^3$, which was lower than the one at the low mountain range site HPB (0.31 $\mu g/m^3$). This result further strengthens our confidence that the low concentrations of precursors are the main cause of the low NPF occurrence

frequency. Accordingly, Figure R3 has been added as Fig.4(c) in the text and the following discussion has been added in Sect.3.1 "NPF occurrence frequency":

"A previous study by Nieminen et al. (2018) found similar annual and seasonal frequencies for MEL and HPB. It is interesting that lower occurrence frequency of NPF is found at ZSF than the two low mountain range sites. The atmosphere in high altitude areas can be influenced by both PBL and FT (Sun et al., 2021; Herrmann et al., 2015; Rose et al., 2017). And NPF was found to be strongly associated with the air parcel vertically transported from lower altitudes (Bianchi et al., 2016; Shen et al., 2016; Tröstl et al., 2016). The influence of vertical transport of PBL air mass is much weaker at ZSF than lower altitudes, leading to lower condensation sink (CS) and concentrations of precursors. As reported in Flentje et al. (2010), the median $SO_2$ mass concentration during the year 2000−2007 at ZSF was about 0.18 μg/m$^3$, which was lower than the one at HPB (0.31 μg/m$^3$). Therefore, though the low temperature and CS at ZSF favour the NPF, the extremely low concentration of precursors at ZSF inhibits the occurrence of NPF, which is likely to be one of the possible reasons of the lower NPF occurrence frequency at ZSF."

[Figure]

**Figure R3: The statistics of condensation sink for NPF days at each GUAN site. The dot indicates the median value, while the line in the middle of box indicates the mean value. The whiskers and lower/upper boundary represent the 90$^{th}$, 75$^{th}$, 25$^{th}$, and 10$^{th}$ percentiles.**

3. Line 286. The authors acknowledge that starting times are influenced by different lower cut points for the measurement sites. This could have been harmonised by extrapolation of data back to 2nm diameter, which is arguably more realistic than using sizes closer to 10nm.

Response:

Thanks for the suggestions. We have extrapolated the starting time from 10 nm to 2 nm with the following equation:

$$t_{2\,nm} = t_{10\,nm} - \frac{(10-2)}{GR_{10-25}} \qquad (1)$$

where $t_{2nm}$ and $t_{10nm}$ are starting time at 2 nm and 10 nm, respectively. The hypothesis behind this extrapolation is that $GR_{10\text{-}25}$ equals to $GR_{2\text{-}25}$. To verify the effectiveness of this conversion, we took MEL as an example (PNSD measured down to 5 nm) and compared $GR_{10\text{-}25}$ and $GR_{5\text{-}25}$ in the year of 2012. The relative difference between $GR_{5\text{-}25}$ and $GR_{10\text{-}25}$ is only 1 %, meaning the bias introduced by the extrapolation is negligible. Thus, we have updated the starting time of all the sites with extrapolated $t_{2\,nm}$ in the main text.

[Figure]

**Figure R4: Scatter plot of NPF starting time at 2 nm ($t_{2\,nm}$) on different days of year. The time displayed in the figure is the overall mean starting time. Black solid lines denote the mean value of starting time at each season, and the red and black dash line indicate the sunrise and sunset time, respectively.**

The scatter plot of the modified starting time $t_{2\,nm}$ is shown in Fig.R4. The modified starting time $t_{2\,nm}$ is about 2 hours earlier than $t_{10\,nm}$. We have modified Fig.8 and corresponding text in "Sect.3.3 Starting time of NPF events" as following:

"Figure 8 shows the estimated starting time of class I events as a function of day of year. The starting time was initially estimated based on local time (UTC+1) and further converted to solar time according to the longitude of the sites. The PNSD observations in our dataset initiate from particle sizes from 5 or 10 nm at different sites (Table 1). However, starting time at 10 nm ($t_{10\,nm}$) was not able to describe the actual occurrence time of nucleation. Therefore, $t_{10\,nm}$ has been converted to the critical nucleation diameter of 2 nm ($t_{2\,nm}$) using the $GR_{10\text{-}25}$ values by $t_{2\,nm} = t_{10\,nm} - \frac{(10-2)}{GR_{10-25}}$.

Typically, most NPF events started between 07:30 and 9:00 solar time at all GUAN sites. Seasonal variations in starting time were evident, with earlier starting time in summer due to earlier sunrise. It is noteworthy that the differences in the starting time of NPF events exist between sites, as shown in Fig.8 and Fig.S7 in the supplementary material. The three mountainous sites (HPB, SCH, and ZSF) had the latest starting time around 09:00. Two UB sites LTR and LWE had the earliest starting time around 07:30. Starting time at BOS and three RB sites (MEL, WAL, and NEU) is around 08:30. Since the use of solar time has already eliminated the bias of local time relative to site longitude, the difference of starting time among sites mainly stems from the different diurnal variation of precursor concentration and CS."

4. Line 177. The method of estimating the effect of NPF upon CCN concentrations appears to be flawed. The method is cited as following two published papers, especially that of Ren et al. (2021). The other reference, Kalkavouras et al. (2019) is incorrect in the reference list, but the correct paper is concerned primarily with kappa values, not this method of estimating CCN enhancements. The clear implication of presenting the CCN enhancement is that the new particles are growing to CCN sizes, which at 4nm/h would take from 12 – 48 hours dependent upon the supersaturation. Taking a ratio of estimated CCN within and after the NPF event to CCN in the two hours before the event is reflecting the growth of particles which were already approaching CCN size as opposed to newly formed particles. The authors may have been aware of this, but it is certainly not made clear to the reader.

Response:

Thanks for the comment. We have revised the reference Kalkavouras et al. (2019) in the reference list as:

Kalkavouras, P., Bougiatioti, A., Kalivitis, N., Stavroulas, I., Tombrou, M., Nenes, A., and Mihalopoulos, N.: Regional new particle formation as modulators of cloud condensation nuclei and cloud droplet number in the eastern Mediterranean, Atmos. Chem. Phys., 19, 6185-6203, doi:10.5194/acp-19-6185-2019, 2019

In this article, they developed an approach for CCN enhancement estimation in Sect.3.2 "Characteristics and interpretation of the Finokalia NPF events", and introduced the details of this method and evaluated the variation of $t_{start}$, $t_{end}$, and the average of CCN number concentration enhancement for different supersaturations.

We agree with the reviewer that both the newly formed particles and pre-existing particles can grow to CCN-relevant size during NPF events. The pre-existing particles have larger diameters and may reach CCN-relevant size faster than the newly formed particles, therefore may even have a larger contribution to CCN number concentration.

Kalkavouras et al. (2019) stated that the pre-existing particles may induce a bias in the estimated CCN enhancement up to 50 %. Following your suggestion, we have evaluated the duration ($t_{end}$-$t_{start}$ in Sect.2.3.5) of NPF contributing on $N_{CCN}$ at MEL. The median duration is 11.2 hour, implying that the pre-existing particles do contribute on $E_{Nccn}$ estimated in this study. However, it is indeed difficult to decompose the contributions of the two parts. To clarify this point, we have added the following discussion in Sect. 4.2:

"To evaluate the potential contribution of NPF to CCN, the relative enhancement of CCN number concentration ($N_{CCN}$ enhancement, denoted as $E_{Nccn}$), which is the ratio between $N_{CCN}$ after and prior to the NPF event, has been estimated following the approach proposed by previous studies (Kalkavouras et al., 2019; Ren et al., 2021). It should be noted that during a NPF event, both the newly formed particles and pre-existing particles can grow to CCN-relevant size. The pre-existing particles have larger diameters and may reach CCN-relevant size faster than newly formed particles, therefore may even have a larger contribution to CCN number concentration. Kalkavouras et al. (2019) stated that the pre-existing particles may induce a bias in the estimated CCN enhancement up to 50 %. It is however difficult to decompose the contributions of the two parts. So the $E_{Nccn}$ estimated in this study was an integrated CCN number concentration enhancement contributed by both the two parts during NPF events. "

5. The authors might also like to consider:

- The abstract states formation rates and growth rates but doesn't state at what diameter. It would be useful to change $J_{nuc}$ and $GR_{nuc}$ in the text to $J_{10\text{-}25}$ and $GR_{10\text{-}25}$ as it is currently confusing.

  Response:

  Thanks for the suggestion. We have replaced $GR_{nuc}$ and $J_{nuc}$ in the manuscript as $GR_{10\text{-}25}$ and $J_{10\text{-}25}$, respectively. The size range of GR and $J$ in the abstract has been clarified as well:

  "The annual mean growth rate between 10 and 25 nm varied from 3.7 to 4.7 nm h$^{-1}$, while the formation rate with same size range 10−25 nm from 0.4 to 2.9 cm$^{-3}$ s$^{-1}$."

- The statement in the introduction "Existing theories still cannot fully explain the fundamental chemical mechanisms of NPF events observed under diverse tropospheric environments and the result of field measurements are often controversial concerning the contribution of the chemical species to nucleation and growth of nanoparticles (Lee et al., 2019)" is not really true – I am not sure that the results of field campaigns are ever controversial, and our current understanding explains *most* of what we observe.

Response:

Thanks for the comment. We have revised the corresponding sentences as "To understand the characteristics of NPF and its influencing factors, field campaign experiments covering a wide range of atmospheric conditions and environments are essential (Lee et al., 2019)".

- There are differences between Leipzig-TROPOS and Leipzig-WEST despite them being very close to one another. NPF starts at the same time, but the frequencies, growth rates, and formation rates (as well as their seasonal trends) differ. Do the authors have any comments on this?

Response:

Thanks for the comment. One possible explanation for such differences between LTR and LWE may be the different surroundings of these two sites. Although both sites represent urban background condition of Leipzig, LTR is relatively more affected by traffic emissions. The LTR is located on the top of a three-floor building about 100 m from the main road. The LWE is located in a park with 30 m distance from a minor road, so that the traffic impact is negligible (Birmili et al., 2016).

Figure R3 shows the statistics of CS for all GUAN sites. It can be seen that CS at LTR is slightly higher than that at LWE. Furthermore, we have compared the PNSD of the two sites for NPF days, undefined event days, and non-NPF days, respectively in Fig.R5. It can be seen that the particle number concentration in the range of 20-60 nm of LTR was higher than LEW during morning and evening rush hours, indicating a greater influence of fresh traffic emission at LTR. In one of our previous studies, we found the particle number concentration of LTR was higher than that of LWE, especially for traffic related size range $N_{10-30}$ and $N_{30-200}$, with 10% and 17% higher, respectively (Sun et al., 2009). Therefore, we can conclude that LTR is more affected by traffic emissions than LWE, which may cause different characteristics of NPF.

[Figure]

**Figure R5: Mean PNSD with respect to NPF days/undefined days/non-event days for LTR and LWE.**

Accordingly, the discussion in Sect.3.2 "Growth and formation rates" has been revised as:

"Additionally, one should be noticed that both LTR and LWE are located in the urban background of Leipzig. The occurrence frequency and starting time of NPF (Sect.3.3) were similar at the two sites, while the $GR_{10-25}$ and $J_{10-25}$ were not. One possible explanation for such differences on $GR_{10-25}$ and $J_{10-25}$ may be the different surroundings of the two sites. LTR is located on the top of a three-floor building about 100 m from a main road, therefore is relatively more influenced by traffic emissions. The LWE is located in the park with 30 m distance from a minor road, so the impact of fresh traffic emission is negligible (Birmili et al., 2016). In one of our previous studies, it was found that the particle number concentration at LTR was higher than that at LWE, especially for traffic related size range $N_{10-30}$ and $N_{30-200}$, with 10 % and 17 % higher, respectively (Sun et al., 2009), indicating higher gaseous precursor concentration and thus stronger anthropogenic influence at LTR."

- It would be very useful to see the condensation sink plotted for each of these sites in the same way that GR and J are plotted.

Response:

We have calculated the condensation sink (CS) for each site. The plot has been added in Fig.4(c), and the corresponding discussions have been added in Sect.3.2 "Growth and formation rates ":

"Figure 4 shows the basic statistics of annual $GR_{10-25}$, $J_{10-25}$ and CS at the nine GUAN sites. The growth and formation rate were only evaluated for class I event in this study, and the CS was estimated for all NPF event days. … The CS values were generally higher in the area with stronger anthropogenic emissions, and the lowest at high Alpine site ZSF. The CS and $J_{10-25}$ at BOS were lower than the other two UB sites in Leipzig, suggesting relatively fewer anthropogenic emissions at BOS than LTR and LWE…"

[Figure]

**Figure 4: Basic statistics of $GR_{10-25}$, $J_{10-25}$ and condensation sink measured at the GUAN sites. Dots denote the mean vales, and the boxes and whiskers denote the 10th, 25th, 50th, 75th, and 90th percentiles.**

- It would also be nice to see the size distributions plotted up. Perhaps a mean size distribution for each site on each event day/non-event day/undefined day. In the supplement, a mean contour plot on an NPF, undefined, and non-event day would be useful to see for each site too.

Response:

The mean PNSD and contour plot with respect to NPF day, undefined day, and non-event days were shown in Fig.R6 and R7, respectively. The figures have been also added as Fig.S5 and S6 in the supplementary material.

[Figure]

**Figure R6: Mean PNSD with respect to NPF days/undefined days/non-event days for each GUAN site.**

[Figure]

**Figure R7: Mean PNSD with respect to NPF days/undefined days/non-event days for each GUAN site.**

---

## Author Response (AR2)

**Response to Reviewer #1:**

**General comments:**

The authors addressed all reviewer comments. The article can be published as is. However, after checking the reviews, I have identified 3 minor technical fixes.

Response:

We appreciated reviewer's positive feedback and suggestions. The point-to-point responses to the reviewer's comments are listed below.

1. Line 249-250. A reference is needed for "The influence of vertical transport of the PBL air mass is much weaker at ZSF than at lower altitudes..." I suggest Collaud-Cohen et al., Atmos. Chemistry. Phys., 18, 12289–12313 (doi:10.5194/acp-18-12289-2018, 2018)

   Response:

   Thanks for the suggestion. We have added the reference Collaud-Cohen et al. (2018) in the main text and the reference list.

   Collaud-Coen, M., Andrews, E., Aliaga, D., Andrade, M., Angelov, H., Bukowiecki, N., Ealo, M., Fialho, P., Flentje, H., Hallar, A. G., Hooda, R., Kalapov, I., Krejci, R., Lin, N. H., Marinoni, A., Ming, J., Nguyen, N. A., Pandolfi, M., Pont, V., Ries, L., Rodríguez, S., Schauer, G., Sellegri, K., Sharma, S., Sun, J., Tunved, P., Velasquez, P., and Ruffieux, D.: Identification of topographic features influencing aerosol observations at high altitude stations, Atmos. Chem. Phys., 18, 12289-12313, doi: 10.5194/acp-18-12289-2018, 2018.

2. Caption to Figure 8: please, replace "red and black dash line indicate..." with "red and black dashed lines indicate..."

   Response:

   Thanks. We have corrected the caption of Fig.8 as:

   "Figure 8: Scatter plot of NPF starting time (solar time) on different days of year. Black solid lines denote the mean seasonal starting time, the red and black dashed line indicate the sunrise and sunset time, respectively."

3. Figures S5 and S6 were added following the reviewer's recommendations; however, these figures are not referred to in the text. Please add a reference to these figures in section 2.2 or in the results section. It is worth highlighting the fine distribution of PNSD for non-events in LTR and LWE; Is there any explanation for this?

Response:

Thanks for the comment. The reference to the Fog.S5 and S6 have been added in Sect. 3.2 and 4.1.1.

The fine distribution of PNSD on non-event for site LTR and LWE shows different influences of traffic emission at the two sites. We have compared the traffic influence at LTR and LWE and added discussion in Sect. 3.2 as follows.

● Revision in Sect. 3.2 Growth and formation rates

"The site-to-site difference in anthropogenic influences can be clearly seen from the mean PNSD on non-event days, as shown in Fig.S5 and S6 in the supplementary material. Similarly, the CS values were generally higher in the area with stronger anthropogenic emissions, and the lowest at high Alpine site ZSF. … . As shown in Fig.S6 (a3) and (b3), the particle number concentration lower than 50 nm at LTR was higher than that at LWE. One of our previous studies showed that the particle number concentration in traffic related size range $N_{10-30}$ and $N_{30-200}$ were 10 % and 17 % higher, respectively at LTR and LWE (Sun et al., 2009), indicating higher gaseous precursor concentration and thus stronger anthropogenic influence at LTR."

● Revision in Sect. 4.1.1 Nucleation strength factor

"…In addition, the mean PNSD on NPF and non-event days for each GUAN site in Fig.S6 can clearly depict this site-to-site difference in NSF. The influence of anthropogenic emissions on UFP gradually decreased from urban background to high Alpine site, leading to clearer background atmosphere for regional background and mountainous area. Hence, the contribution of NPF on UFP were more pronounced in these site categories."